# Mechanistic insights into global suppressors of protein folding defects

**Gopinath Chattopadhyay**[1☯]**, Jayantika Bhowmick**[1☯]**, Kavyashree Manjunath**[2]**,
Shahbaz Ahmed**[1]**, Parveen Goyal**[3]**, Raghavan Varadarajan**[1]*

**1** Molecular Biophysics Unit, Indian Institute of Science, Bangalore, India, **2** Centre for Chemical Biology and Therapeutics, Institute For Stem Cell Science and Regenerative Medicine, Bangalore, India, **3** Institute for Stem Cell Science and Regenerative Medicine, Bangalore, India

☯ These authors contributed equally to this work.
* varadar@iisc.ac.in

**Data Availability Statement:** Crystal structures of the CcdB mutants have been deposited in Research Collaboratory for Structural Bioinformatics Protein Data Bank (RCSB PDB) with PDB IDs: 7EPG (CcdB

## Abstract

Most amino acid substitutions in a protein either lead to partial loss-of-function or are near neutral. Several studies have shown the existence of second-site mutations that can rescue defects caused by diverse loss-of-function mutations. Such global suppressor mutations are key drivers of protein evolution. However, the mechanisms responsible for such suppression remain poorly understood. To address this, we characterized multiple suppressor mutations both in isolation and in combination with inactive mutants. We examined six global suppressors of the bacterial toxin CcdB, the known M182T global suppressor of TEM-1 β-lactamase, the N239Y global suppressor of p53-DBD and three suppressors of the SARS-CoV-2 spike Receptor Binding Domain. When coupled to inactive mutants, they promote increased *in-vivo* solubilities as well as regain-of-function phenotypes. In the case of CcdB, where novel suppressors were isolated, we determined the crystal structures of three such suppressors to obtain insight into the specific molecular interactions responsible for the observed effects. While most individual suppressors result in small stability enhancements relative to wildtype, which can be combined to yield significant stability increments, thermodynamic stabilisation is neither necessary nor sufficient for suppressor action. Instead, in diverse systems, we observe that individual global suppressors greatly enhance the foldability of buried site mutants, primarily through increase in refolding rate parameters measured *in vitro*. In the crowded intracellular environment, mutations that slow down folding likely facilitate off-pathway aggregation. We suggest that suppressor mutations that accelerate refolding can counteract this, enhancing the yield of properly folded, functional protein *in vivo*.

## Author summary

Global suppressor mutations rescue defects caused by diverse loss-of-function mutations, and are key drivers of protein evolution. However, the mechanisms responsible for such suppression remain poorly understood. Most individual, global suppressors result in small stability enhancements relative to wildtype, which can be combined to yield significant stability increments. We show here that thermodynamic stabilisation is neither necessary nor sufficient for suppressor action. Individual suppressors greatly enhance the

S12G), 7EPJ (CcdB V46L) and 7EPI (CcdB S60E). The validation reports of the structures are submitted along with the manuscript. The PDB doi of the structures are: 7EPG (10.2210/pdb7EPG/pdb); 7EPJ (10.2210/pdb7EPJ/pdb), 7EPI (10.2210/pdb7EPI/pdb). The data relevant to the figures in the paper have been made available within the article and in the supplementary information section.

**Funding:** This work was funded in part by a grant to RV from the Department of Biotechnology (DBT), grant number-BT/COE/34/SP15219/2015, DT.20/11/2015), Government of India (https://dbtindia.gov.in/) and from the Bill and Melinda Gates Foundation (INV-005948) (https://www.gatesfoundation.org/). Funding for infrastructural support was from Department of Science and Technology-Fund for Improvement of S&T Infrastructure (DST-FIST) (https://dst.gov.in/), University Grants Commission (UGC) Centre for Advanced study (https://www.ugc.ac.in/), Ministry of Education (MOE) (https://www.education.gov.in/en), and the DBT IISc Partnership Program. G.C. and J.B. acknowledge Ministry of Education (MOE) for their fellowships. S.A. is thankful to Department of Biotechnology (DBT) (BT/IN/EU-INF/15/RV/19-20) for his research fellowship. K.M. acknowledges Department of Science and Technology-Science and Engineering Research Board (DST–SERB) for financial support, sanction order no: PDF/2017/002641 (http://www.serb.gov.in/home.php). P.G. acknowledges Department of Biotechnology (DBT)/Wellcome Trust India Alliance for fellowship [grant number IA/E/16/1/502999] (https://dbtindia.gov.in/). The funders had no role in study design, data collection and analysis, decision to publish, or preparation of the manuscript.

**Competing interests:** The authors have declared that no competing interests exist.

foldability of folding defective, loss-of-function mutants, primarily through increase in refolding rate constants and burst phase amplitudes. Similar trends are observed in diverse proteins from both prokaryotes and eukaryotes, including cytoplasmic and secreted proteins, confirming the generality of these results.

## Introduction

Correlated mutational data extracted from multiple sequence alignments, can guide structure prediction of a protein [1] or help in the determination of interfacial residues responsible for binding to various partners [2]. Experimental studies have often led to the identification of second suppressor mutations which can alleviate the defects in folding, function and stability of the protein caused by an initial deleterious mutation. Such compensatory mutations, often occur within the same gene, highlighting the role of intragenic suppression in evolution [3].

Suppressors can be either spatially proximal or distal to the inactivating mutation. Distal suppressors, are typically able to suppress multiple inactivating mutations, not necessarily in proximity to each other, and are often referred to as global suppressors. Prior studies have suggested that such suppressors function either by (a) increasing global thermodynamic stability [4], (b) enhancing the activity of the native protein without any effect on stability [5], or (c) improving the folding of the native protein without substantial enhancement of the thermodynamic stability [6]. In laboratory-based evolution experiments, it has been shown that the evolution of new function is accompanied by second-site compensatory mutations, which compensate for the probable destabilizing function altering mutations [7]. Previous studies have primarily focused on thermodynamic rather than kinetic effects of mutations on protein stability and function, though the latter may be more relevant *in vivo* [8].

Here, we endeavour to provide insights into the mechanisms responsible for global suppression. The primary experimental system utilised is a 101-residue homo-dimeric protein, CcdB (Controller of Cell Death protein B), which is a part of the toxin-antitoxin (CcdB-CcdA) module involved in the maintenance of F-plasmid in *Escherichia coli* [9]. We probe effects of multiple global suppressors of CcdB, on the folding kinetics, stability, *in vivo* solubility and *in vivo* activity of the protein. The suppressors are able to rescue folding defects of the inactive mutants, through thermodynamic and kinetic stabilization, with the largest effects on refolding rates. To understand the structural basis of stabilization, crystal structures of three CcdB suppressor mutants, namely, S12G, V46L and S60E, were solved. We probed the effects of the suppressor mutants on aggregation, binding and thermal tolerance. We also examined the effects on stability and folding of two known global suppressors: M182T of TEM-1 β-lactamase, an extended spectrum β-lactamase (ESBL) enzyme conferring antibiotic resistance against third generation cephalosporins and N239Y in the DNA binding domain (DBD) of p53, a critical tumour suppressor protein, which is known to suppress the effect of oncogenic inactive mutants. We additionally examined similar parameters for recently isolated suppressor mutations in RBD (Receptor Binding Domain of spike glycoprotein of SARS-CoV2), [10]. The analysis of data for these diverse systems provided general insights into the mechanism of action of global suppressors.

## Results

### Phenotypic characterisation of putative second-site suppressor mutations in CcdB

Previous characterisation of a single site saturation mutagenesis (SSM) library of ~1600 CcdB mutants led to the identification of five partial (or) complete loss-of-function mutants, varying

in their substitutional patterns of sizes and polarities, and were termed as parent inactive mutants (PIMs), (namely,V5F, V18W, V20F, L36A and L83S) [8,11,12]. Second-site saturation mutagenesis libraries, generated by individually incorporating each PIM into the SSM library, were exhaustively screened for suppressors by checking their GyrA14-binding abilities using the technique of yeast surface display (YSD) coupled to FACS [1]. PIMs bound GyrA14 poorly. Two residues, R10 and E11, located on an exposed loop region far from the site of PIMs, were identified as sites for distal suppressors (Fig 1A). The global suppressor R10G rescued Gyr-A14-binding defects at all five PIM positions, and increased the apparent $T_m$ by 8˚C, relative to WT CcdB but E11R was not characterized [1]. Subsequently, another putative global suppressor S12G was identified by analyzing saturation suppressor libraries using FACS coupled to deep sequencing [13]. S12, located beside R10 and E11 on the exposed loop, is involved in hydrogen bonding with the cognate antitoxin CcdA (Fig 1A). Phenotypes of the E11R and

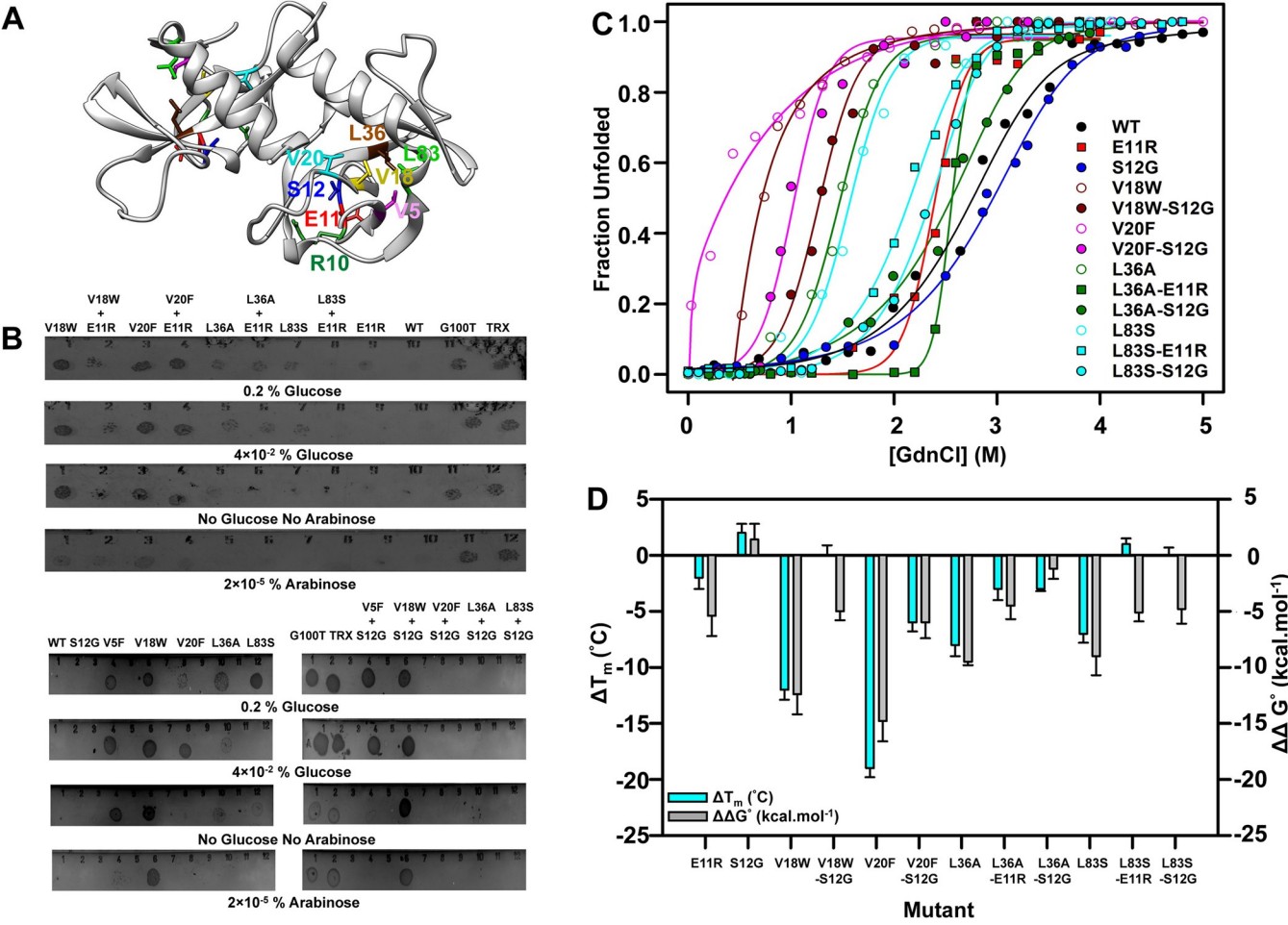

**Fig 1. Folding defects rescued by suppressors.** (A) Experimentally obtained inactive mutants and distal suppressors mapped onto the crystal structure of dimeric CcdB (PDB ID 3VUB [71]). The distal suppressors at R10, E11and S12, are mapped on an exposed loop region while the PIMs at V5, V18, V20, L36 and L83 are present in the core of the protein. (B) *In-vivo* activity of the PIMs at 37˚C in the background of the suppressors E11R and S12G. The condition where growth ceased was reported as the active condition. Four representative conditions are shown. G100T, an inactive CcdB mutant, and thioredoxin (TRX) are used as controls, which grow in all the repressor/inducer conditions. (C) Equilibrium GdnCl denaturation profiles using nanoDSF are shown. The experimental data are shown in symbols, while the fits are shown in solid lines. (D) The difference in thermal stability, $\Delta T_m$ ($\Delta T_m = T_{m_{Mutant}} - T_{m_{WT}}$) (in cyan), and thermodynamic stability assayed by chemical denaturation, $\Delta\Delta G˚$ ($\Delta\Delta G˚ = \Delta G˚_{Mutant} - \Delta G˚_{WT}$) (in grey) of the different CcdB mutants. The error bars wherever shown represent the standard deviation from two independent experiments, each performed in duplicates. Numerical values are listed in S1 and S2 Tables.

S12G CcdB mutants were studied here after cloning the mutants both individually and in combination with PIMs into the *E. coli* expression vector pBAD24, under the control of the $P_{BAD}$ promoter [14]. This allows for tuneable expression with glucose (repressor) and arabinose (inducer). The plasmids were individually transformed into the CcdB-sensitive *E.coli Top10*p-JAT strain. The cells were grown at different concentrations of glucose and arabinose [8,12] to allow increasing levels of CcdB expression. Mutant phenotypes were studied as a function of varying repressor (glucose) and inducer (arabinose) concentrations. For WT and the fully active mutants E11R and S12G, the cells fail to grow even at the highest glucose (repressor) concentration (0.2% glucose), because even very low levels of active CcdB result in cell death. The inactive mutants grow even at higher concentrations of arabinose ($7\times10^{-5}$% arabinose). In the background of the E11R or S12G suppressor mutations, with the exception of V18W-S12G, the WT-like phenotype of the remaining (PIM, suppressor) pairs is restored at lower glucose concentrations ($\leq$0.2%), leading to cell death (Fig 1B).

To examine the relative *in vivo* solubility levels (proxy for folded functional forms of proteins), *E. coli* strain *Top10GyrA* was individually transformed with each mutant. The solubilities of the PIMs were significantly lower than that of WT CcdB and suppressors E11R and S12G (S1 Table). However, the solubilities of the inactive mutants in the background of the suppressors were significantly enhanced (S1A Fig). These results reveal that lowered activity and decrease in solubility of the inactive proteins are both improved in the background of the suppressors.

## Enhancement of thermal and chemical stability of multiple inactive mutants by global suppressors

The purified proteins (4 μM) were subjected to thermal denaturation using nanoDSF and the apparent $T_m$ was calculated (Fig 1D). S12G had a 2˚C higher $T_m$, indicating that the mutation is stabilising (S1 Table). Further the inactive mutant-suppressor pairs showed increased apparent thermal stabilities (~5–12˚C) relative to the inactive mutants. The ability of the purified proteins to bind to CcdA peptide (8 μM) was also examined by monitoring thermal denaturation using binding of Sypro orange dye [15], both in the absence and presence of CcdA peptide (45–72). Relative to the free proteins, apparent $T_m$'s of the CcdA-bound complexes showed increments (S1B Fig) due to stability-enhancements of the proteins in the presence of peptide [1].

Equilibrium unfolding experiments for the CcdB mutants were carried out by nanoDSF. The data were fitted to $N_2\leftrightarrow2D$ unfolding models for homo-dimeric CcdB as described [14,16]. The fraction unfolded of different CcdB mutants (200 mM HEPES, pH 8.4) in the presence of GdnCl is plotted as a function of denaturant concentration (Fig 1C). The midpoint of chemical denaturation ($C_m$), $\Delta G^0$ and m-values were measured for all the CcdB mutants (S2 Table). The suppressor S12G alone is 0.8 kcal/mol more stable than the WT, and the double mutants are apparently 2.5–5.0 kcal/mol more stable than the inactive mutants (Fig 1D). The significant difference in the apparent effect of the suppressor in the context of PIM relative to WT, is likely due to the high tendency of inactive mutants to aggregate over time thereby reducing the amount of functionally folded form. Surprisingly, E11R is 3.1 kcal/mol less stable than WT as assayed by chemical denaturation studies and has a $T_m$ 2˚C lower than WT but still acts as a global suppressor.

## Suppressor substitutions accelerate the refolding rate and reduce the unfolding rate of the WT and multiple inactive mutants

Refolding and unfolding kinetics for CcdB mutants were also monitored by time-course fluorescence spectroscopy at 25˚C using nanoDSF [16]. Refolding was performed at pH 8.4, at

final GdnCl concentrations ranging from 0.5 M-1.5 M. During refolding, the two monomers rapidly associate in a diffusion limited process [17]. Refolding for the WT occurs with a fast and a slow phase as observed earlier [17]. The suppressors E11R and S12G refold at faster rates compared to the WT. Relative to the WT, the inactive mutants refold slowly, whereas their refolding rates were enhanced (both fast and slow phase) in the background of the suppressor (Fig 2A, S3 Table). The fast phase of refolding for L83S-E11R could not be captured owing to its fast refolding kinetics. The noise associated with refolding kinetics of L36A-S12G in the presence of 1 M GdnCl is due to the faster kinetics of refolding and high dead time of about ~15 seconds associated with the instrument.

The unfolding trace of WT CcdB when fitted to the three-parameter unfolding equation, gives a fitted unfolding rate of 0.05 s$^{-1}$. S12G shows a much slower rate of unfolding. For all the PIMs, we observed very fast unfolding even at low GdnCl concentrations, whereas the mutant-suppressor pairs had slower unfolding rates relative to the individual PIMs, though typically faster than the rates for WT and the suppressors E11R and S12G (Fig 2B, S3 Table). Refolding and unfolding reactions were carried out at three different GdnCl concentrations and the observed rate constants and m values were plotted as a function of GdnCl concentration (S2 Fig, S4 Table). Using this, the refolding rate constants for both fast and slow phases, and unfolding rate constants were calculated at 0 M GdnCl as described previously [16] for relative comparison (S4 Table). These experiments correlate with other results and indicate that the suppressors E11R and S12G have lower to marginally higher thermal stabilities than WT and that all the mutant-suppressor pairs are both kinetically and thermodynamically more stable than the corresponding PIMs.

## Rescue of activity and stability of refolded CcdB proteins by the global suppressors

Microscale thermophoresis (MST) was used to measure the binding affinities of fluorescently labeled GyrA14 (used at a fixed concentration of 70 nM) to unlabelled CcdB mutant proteins (native, native in GdnCl and refolded proteins). The obtained data indicates that WT CcdB, S12G and E11R bind GyrA14 with $K_D$'s of about 2.2, 2.3 and 1.2 nM respectively which is consistent with the SPR binding studies (Figs 2C and S3A and S1 Table). Labeled GyrA14 also bound with similar affinity to the native and refolded proteins in 1.5 M GdnCl (Figs 2C and S3B–S3C and S1 Table), indicating that refolding was reversible. The $C_m$ of GyrA14 was determined to be 4.48 M, confirming that it was folded at 1.5 M GdnCl used in the above refolding assay (S3D Fig).

The refolded CcdB proteins and the native proteins in the presence of 1.5 M GdnCl were also subjected to thermal denaturation, and the apparent $T_m$ was calculated (Fig 2D, S2 Table). The near identicality of the $T_m$'s for native and refolded proteins, further confirms that the mutants refold reversibly. Except for the PIMs V18W and V20F, all the other mutants showed clear thermal transitions confirming that they were in a folded conformation in the presence of GdnCl. The suppressors improved the $T_m$ of the refolded proteins, relative to the PIMs.

## Enhancement of thermal tolerance of the CcdB mutants by the global suppressors

The binding of purified CcdB mutants to their target GyrA14, was also probed using SPR. WT CcdB, S12G and E11R bind to GyrA14 with $K_D$'s of about 1.4, 2.6 and 2.5 nM respectively (S4 Fig, S1 Table). An increased affinity for DNA Gyrase for the inactive mutant-suppressor pairs was observed in all the cases, consistent with functional rescue. The apparent low affinity of

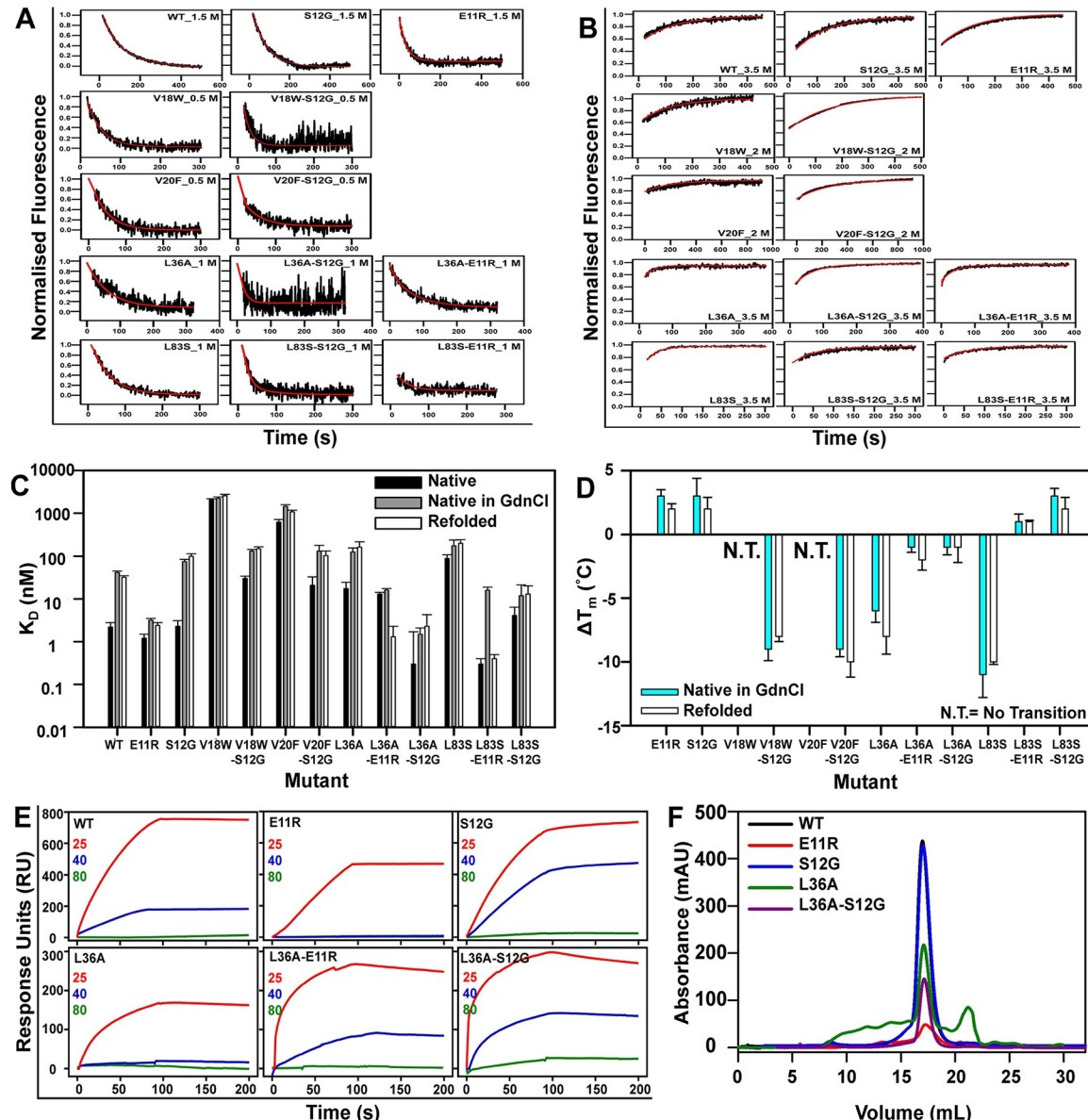

**Fig 2. Kinetic and thermodynamic stabilisation by the CcdB suppressor mutants.** (A) CcdB proteins exhibit biphasic refolding kinetics with a fast and slow phase whereas (B) unfolding of CcdB proteins follows single exponential kinetics. The experimental kinetic traces obtained at different GdnCl concentrations are shown in black, while the fits are shown in red. (C) Interaction between native (black), native protein in GdnCl (grey) and refolded (white) CcdB mutant proteins and labeled GyrA14 analyzed by MST. (D) The difference in apparent thermal melting temperatures ($\Delta T_m = T_{m_{Mutant}} - T_{m_{WT}}$) for native proteins in GdnCl (cyan), and refolded proteins (grey). The error bars wherever shown represent the standard deviation from two independent experiments, each performed in duplicates. (E) Binding of 500 nM WT and mutant CcdB to immobilised GyrA14 measured by passing the same concentrations of the analyte (CcdB proteins), after heat stress at two different temperature (40 and 80°C), followed by cooling back to 25°C. A room temperature control (25°C) was also used. The residual active fraction was calculated as described in the materials section. (F) The SEC profiles of a few of the CcdB mutants are shown. The PIM L36A shows aggregation as well as degradation as compared to the WT, and E11R and S12G suppressors. The L36A-S12G has a similar profile like the WT and S12G. Kinetic parameters from the fits are listed in S3 Table and values extrapolated to zero denaturant are listed in S4 Table.

these inactive mutants may also arise due to the inability to correctly estimate the fraction of active protein for these purified mutants. The SEC profile of the PIM L36A, shows a significant amount of aggregation as well as degradation as compared to the WT, suppressor S12G and

the double mutant L36A-S12G (Fig 2F). All the SEC experiments were carried out using 100 μg of protein at a flow rate of 0.5 mL/min. For the other mutants, due to poor yields and high tendency to aggregate, SEC was not performed. Thermal tolerance of E11R, S12G, L36A, L36A-E11R, L36A-S12G and WT CcdB, was also assessed by determining the binding of CcdB proteins to GyrA14 after prolonged heat stress followed by cooling to room temperature (Fig 2E, S2 Table). The S12G mutant retained 4% activity after incubation at 80°C for 1 hour, representing a five-fold improvement over WT, whereas the double mutants L36A-E11R and L36A-S12G showed three-fold and ten-fold improvement over WT respectively. Surprisingly, L36A-E11R had residual activity at 40°C, as compared to E11R alone which lost activity at this temperature.

## Thermodynamic and kinetic stabilisation mediated by the M182T global suppressor substitution in extended spectrum TEM-1 β-lactamases conferring antibiotic resistance

TEM-1 β-lactamase, a 263 residue monomeric enzyme confers resistance to β-lactam antibiotics [18]. There have been several studies on its stability and folding kinetics making it a suitable system to study the effect of global suppressor substitutions [19,20]. Located far from the active site, M182T, a drug-resistant clinically isolated mutation showing extended spectrum β-lactamase (ESBL) activity, increases protein expression and restores stability defects caused by active-site substitutions [21]. M182T rescues a folding-defective M69I mutant thereby conferring resistance to inhibitors such as clavulanate (Inhibitor-resistant TEM β-lactamases, IRTs), and a core engineered substitution L76N, where L76 has been shown to be sensitive to substitutions [6,22] with M182T being distant from the primary mutation (Fig 3A).

In the present study, we characterized the global suppressor M182T and observed its effect on stability and folding of known inactive mutants (Fig 3). We determined the MIC and $IC_{90}$ of the TEM-1 WT, M182T, M69I, M69I-M182T, L76N and L76N-M182T for both ampicillin and cefotaxime, a third-generation cephalosporin (Fig 3B and 3C, S5 Table). In line with a previous study, we found that the M182T suppressor alone and the M69I-M182T double mutant have comparable values to that of the WT [23] and M69I respectively [24], however M182T rescues the activity of the inactive enzyme mutant L76N (Fig 3B and 3C, S5 Table) [22,25]. Further the activity of the purified mutants was monitored *in vitro* with nitrocefin (Fig 3D). The results obtained showed similar effects of the M182T substitution on WT or M69I or L76N background as observed *in vivo* (Figs 3D, S5C–S5E).

The M182T suppressor substitution enhanced both the thermal and chemical stability of the WT protein as well as the inactive mutants M69I and L76N (Fig 3E and 3F, S5 Table), in agreement with previously published results [26]. When compared with WT, M69I and L76N are less stable (Fig 3G). The suppressor alone is 6°C, 1.8 kcal/mol more stable than the WT, and M69I-M182T and L76N-M182T are 7°C, 2 kcal/mol and 8°C, 1.7 kcal/mol more stable than the inactive mutants M69I and L76N respectively (Fig 3G).

Refolding (in 0.5 M GdnCl) and unfolding (in 2.5 M GdnCl) kinetics for all the mutants (5 μM) were also monitored using nanoDSF. M182T shows a slightly slower rate of unfolding whereas the inactive mutants M69I and L76N, had two and three fold faster unfolding rates respectively as compared to the WT. The mutant-suppressor pairs however had a slower rate of unfolding as compared to the individual inactive mutants (Figs 3H and S5B and S5 Table). The refolding rate constants for each of these mutants, were also calculated for both fast and slow phases. The M182T suppressor alone, refolds at a faster rate compared to the WT. The inactive mutants showed slower refolding kinetics than the WT. In the background of the

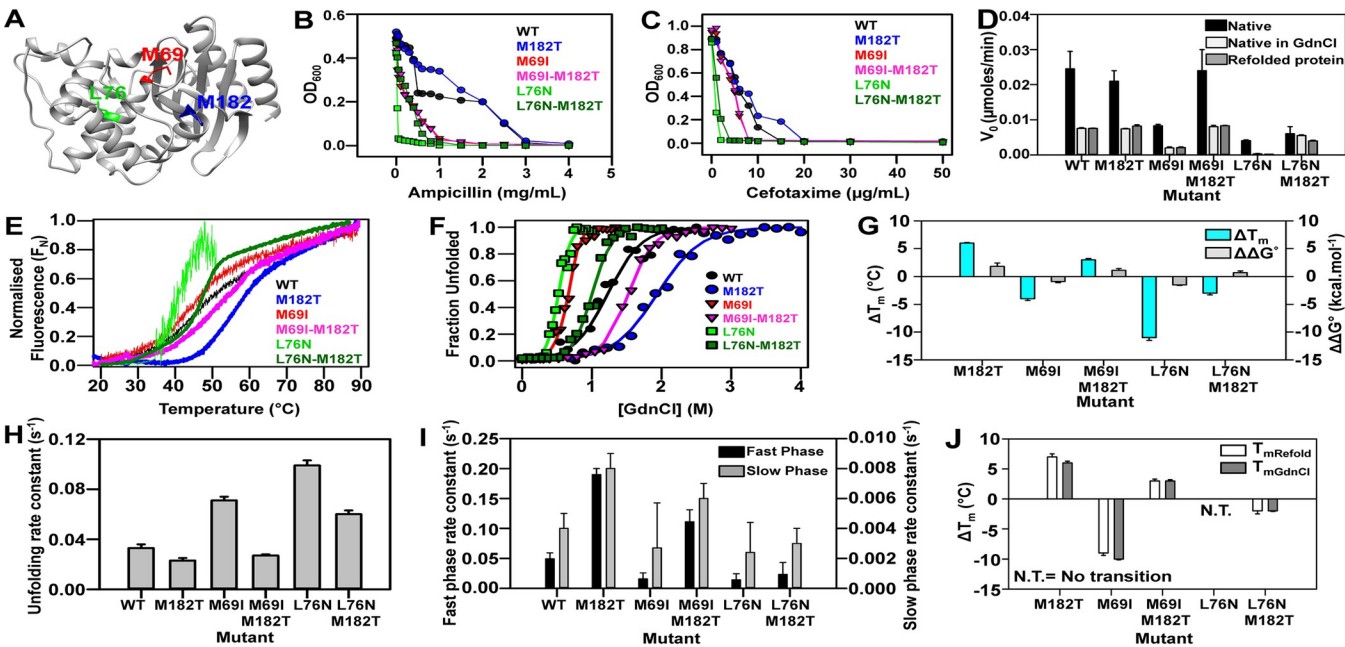

**Fig 3. Enhancement of protein stability by M182T global suppressor in extended spectrum TEM-1 β-lactamases.** (A) Inactive mutants and distal suppressor M182T mapped onto the crystal structure of TEM-1 (PDB ID 1XPB [76]). The TEM-1 protein is shown in ribbon with the distal suppressor M182T, mapped on an exposed region, while the inactive mutants L76N and M69I are present in the core of the protein. (B-C) Influence of M182T substitution on TEM-1 mediated MIC levels using ampicillin (B) and cefotaxime (C) broth dilutions and pET24a plasmid. (D) The initial velocity ($V_0$) of the native enzyme (black), protein in 0.5 M GdnCl (light grey) and refolded protein in 0.5 M GdnCl (dark grey) at 25°C with 10 nM protein and 50 µM of nitrocefin. (E-F) Thermal unfolding profiles and equilibrium GdnCl denaturation profiles of 10 µM of purified WT, and TEM-1 β-lactamase mutants. (G) The difference in thermal stability, $\Delta T_m$ ($\Delta T_m = T_{m_{Mutant}} - T_{m_{WT}}$) of native proteins (in cyan), and thermodynamic stability assayed by chemical denaturation, $\Delta\Delta G°$ ($\Delta\Delta G° = \Delta G°_{Mutant} - \Delta G°_{WT}$) of the different TEM-1 mutants (in grey). (H-I) The observed rate constants of unfolding (2.5 M GdnCl) and the observed rate constants of the fast phase (black) and slow phase (grey) of refolding (0.5 M GdnCl) of different mutants are represented. (J) The difference in thermal stability ($\Delta T_m = T_{m_{Mutant}} - T_{m_{WT}}$) of 10 µM of native proteins (white) and refolded proteins in 0.5 M GdnCl (grey). The error bars wherever shown represent the standard deviation from two independent experiments, each performed in duplicates.

suppressor, however the inactive mutants refold at a faster rate (both fast and slow phase) (Figs 3I and S5A and S5 Table).

The refolded TEM-1 proteins and the native proteins in the presence of 0.5 M GdnCl were also subjected to thermal denaturation, and the apparent $T_m$ was calculated (Fig 3J and S5 Table) which was similar in both the cases. Except for the inactive mutant L76N, all the other mutants showed a proper transition indicating that they were in a folded conformation in the presence of GdnCl and the $T_m$ of the refolded suppressors were also higher than the corresponding inactive mutants (Fig 3J). Further, lactamase activity of native and refolded proteins, both in 0.5 M GdnCl, monitored by nitrocefin hydrolysis yielded results consistent with the other studies (Figs 3D and S5C–S5E).

These experiments indicate that the M182T suppressor alone and in the mutant-suppressor pairs (M69I-M182T and L76N-M182T) rescues the folding defects of inactive mutants and confers higher thermodynamic and kinetic stability relative to the inactive mutants. This is in contrast to a previous study [6] which indicated that the L76N-M182T double mutant was destabilized relative to the L76N inactive mutant, despite the fact that the double mutant showed higher activity *in vivo*. We show that the major contribution to stability mediated by the M182T suppressor substitution is a high refolding rate which allows reversible refolding, even in the background of the inactive mutants.

## Rescue of common oncogenic mutations by an N239Y global suppressor in p53-DBD

~ 50% of human cancers are associated with structurally- or functionally–defective inactive mutations of p53, a transcription factor acting as tumor suppressor, owing to the thermodynamic instability of its core, the DNA Binding Domain (DBD) [27–30]. Previous studies have identified second-site suppressor mutations in the DBD which could restore the WT p53 functionality [31]. One such suppressor mutation, N239Y in the L3 loop of the DBD, could globally rescue multiple missense mutations located at varying regions of the protein by thermodynamic stabilisation of the inactivated core [29,32]. The activity of two of the destabilising oncogenic mutations, located in the core of the DBD, V143A and V157F, were restored by N239Y [27,29,32,33].

In the current study, we aimed to obtain mechanistic insight into the N239Y-mediated suppression of the inactivating p53 mutations, V143A and V157F (Fig 4A). The double mutants V143A-N239Y and V157F-N239Y showed enhanced expression in the soluble fraction *in vivo* in *E. coli*, relative to their corresponding inactive mutants. WT and N239Y single suppressor mutant showed comparable soluble expression levels and yield (Fig 4B). The inactive mutants, V143A and V157F, owing to their low expression levels and aggregation-prone natures, could not be purified or characterised. N239Y, in isolation and in conjunction with the inactive mutants, marginally enhanced the thermal and chemical stabilities of the WT p53 (Fig 4C and 4D, S6 Table). Relative to the WT protein, the suppressor N239Y alone, and the mutants, V143A-N239Y, V157F-N239Y enhance the apparent thermal stabilities by ~1.3°C, 1°C and 3°C respectively. The N239Y, V143A-N239Y and V157F-N239Y have marginal increments in their thermodynamic stabilities over the WT protein in the range of ~ 0.1–0.4 kcal/mol (Fig 4E).

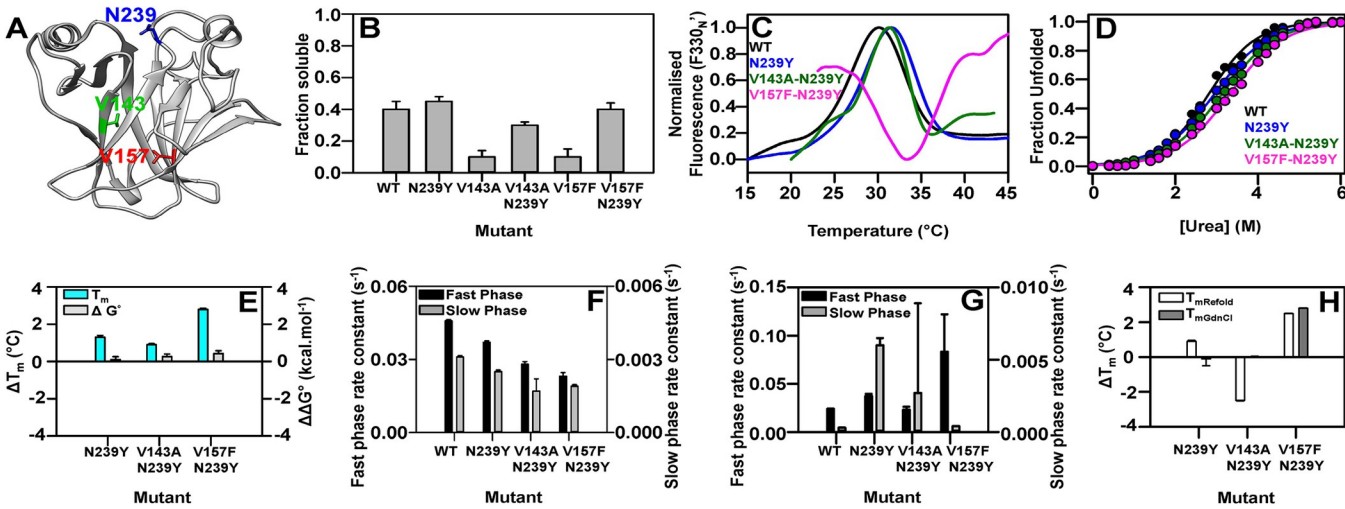

**Fig 4. Characterisation of an N239Y global suppressor in DNA Binding Domain of p53.** (A) Inactive mutants and distal suppressor N239Y mapped onto the crystal structure of p53-DBD (PDB ID 2OCJ [77]). The p53-DBD is shown in ribbon with the suppressor N239Y, mapped on an exposed region, while the inactive mutants V143A and V157F are present in the core of the protein. (B) Influence of N239Y substitution on p53-DBD solubility levels. (C-D) Thermal unfolding profiles and equilibrium Urea denaturation profiles respectively of purified WT and p53-DBD mutants. (E) The difference in thermal stability, $\Delta T_m$ ($\Delta T_m = T_{m_{Mutant}} - T_{m_{WT}}$) of native proteins (in cyan), and thermodynamic stability assayed by chemical denaturation, $\Delta\Delta G°$ ($\Delta\Delta G° = \Delta G°_{Mutant} - \Delta G°_{WT}$) (in grey) of the different p53-DBD mutants. (F-G) The observed rate constants of the fast phase (black) and slow phase (grey) of unfolding in 4.4 M Urea (F) and of refolding in 2 M Urea (G) of different mutants are represented. (H) The difference in thermal stability ($\Delta T_m = T_{m_{Mutant}} - T_{m_{WT}}$) of 10 μM of native proteins (white) and refolded proteins in 0.5 M Urea (grey). The error bars wherever shown represent the standard deviation from two independent experiments, each performed in duplicates.

Refolding (in 2 M urea) and unfolding (in 4.4 M urea) kinetics for the WT and mutants were monitored using nanoDSF at 15˚C. The unfolding traces, yielded comparable slow-phase unfolding rates for the WT and N239Y proteins (0.003 s$^{-1}$ and 0.0025 s$^{-1}$ respectively), whereas the unfolding rates for the double mutants were slightly lower than that for the WT (0.0017 s$^{-1}$ for V143A-N239Y and 0.0019 s$^{-1}$ for V157F-N239Y) (Figs 4F and S5G and S6 Table). The fast-phase unfolding rates for the single and double mutants were marginally lower than that for the WT (S6 Table). The refolding traces, yielded slow-phase refolding rates which were remarkably increased by ~20 and ~10 fold for N239Y and V143A-N239Y respectively, relative to the WT. V157F N239Y refolded with a similar slow-phase rate constant, when compared with the WT. With respect to the fast phase, V157F-N239Y refolds faster than the WT by ~ 3.5 fold, whereas the N239Y and V143A-N239Y refold with similar or marginal increments when compared with the WT (Figs 4G and S5F and S6 Table).

Thermal denaturation of the refolded p53 proteins, along with the native proteins in 0.5 M urea as controls was carried out (Fig 4H). ~ 3˚C increment was observed for the apparent T$_m$ of refolded V157F-N239Y relative to that of the refolded WT, whereas the apparent T$_m$'s for the refolded proteins of N239Y and V143A-N239Y were similar to that for the refolded WT (Fig 4H, S6 Table).

Thus, the N239Y suppressor mutation likely rescues the inactivated destabilised p53 core by marginal enhancement of the thermodynamic and more importantly the kinetic stability of the proteins containing the suppressor, with the largest effect on the refolding rates.

## Effects of global suppressor substitutions in the WT background

In order to further, investigate the role of suppressors on protein stability in the WT background, we performed detailed thermodynamic and kinetic studies of the suppressors alone in CcdB and mRBD proteins (Fig 5). In a recent study employing the PIMs L36A, V18D, V18G and V20G (chosen to span a range of stabilities), several other CcdB suppressor substitutions were also identified using yeast surface display coupled to deep sequencing [13]. In the present study, we selected three such suppressor substitutions, Y8D, V46L and S60E with ΔT$_m$>3˚C (Fig 5A). The purified proteins were subjected to chemical denaturation (Fig 5B). The suppressors Y8D, V46L and the suppressor S60E were ~3 kcal/mol and ~4 kcal/mol respectively more stable than the WT (Fig 5C, S7 Table). The suppressors were also subjected to unfolding (in 4.5 M GdnCl) and refolding (in 2 M GdnCl) kinetic studies. The unfolding rates of the suppressors were 2–2.5 times slower than the WT (Figs 5D and S6B and S8 Table), whereas the fast and slow phase refolding rates of the suppressors were 2–5 times and 9–14 times faster than the WT respectively (Figs 5E and S6A and S8 Table). Further, the proteins were refolded in 1, 2, 3 and 4 M GdnCl and subjected to thermal denaturation. Native protein at the same GdnCl concentrations was used as control (Figs 5F and S6E–S6H). The WT could refold back till 2 M GdnCl, whereas the suppressors Y8D, V46L and suppressor S60E could refold even at 3 and 4 M GdnCl respectively (S6E–S6H Fig). In all cases, the refolded proteins have a broader transition than the native proteins in GdnCl, likely due to the formation of aggregates during refolding.

Next, we investigated the role of the suppressor substitutions in the context of the receptor binding domain (RBD) of SARS-CoV-2 [34]. Using similar saturation suppressor methodology, we recently identified three suppressors of folding defective mutants in this protein [10]. These suppressors, D389E, L390M and P527I are located on the protein surface (Fig 5G). When individually introduced into WT mRBD, they show a ΔT$_m$ of ~3˚C (Fig 5H) [10]. The suppressors were also subjected to unfolding (in 3 M GdnCl) and refolding (in 0.5 M GdnCl) kinetic studies. The unfolding rates of the suppressors were ~4 times slower than the WT for the fast phases and 2 times slower than the WT for slow phases (Figs 5I and S6D and S8

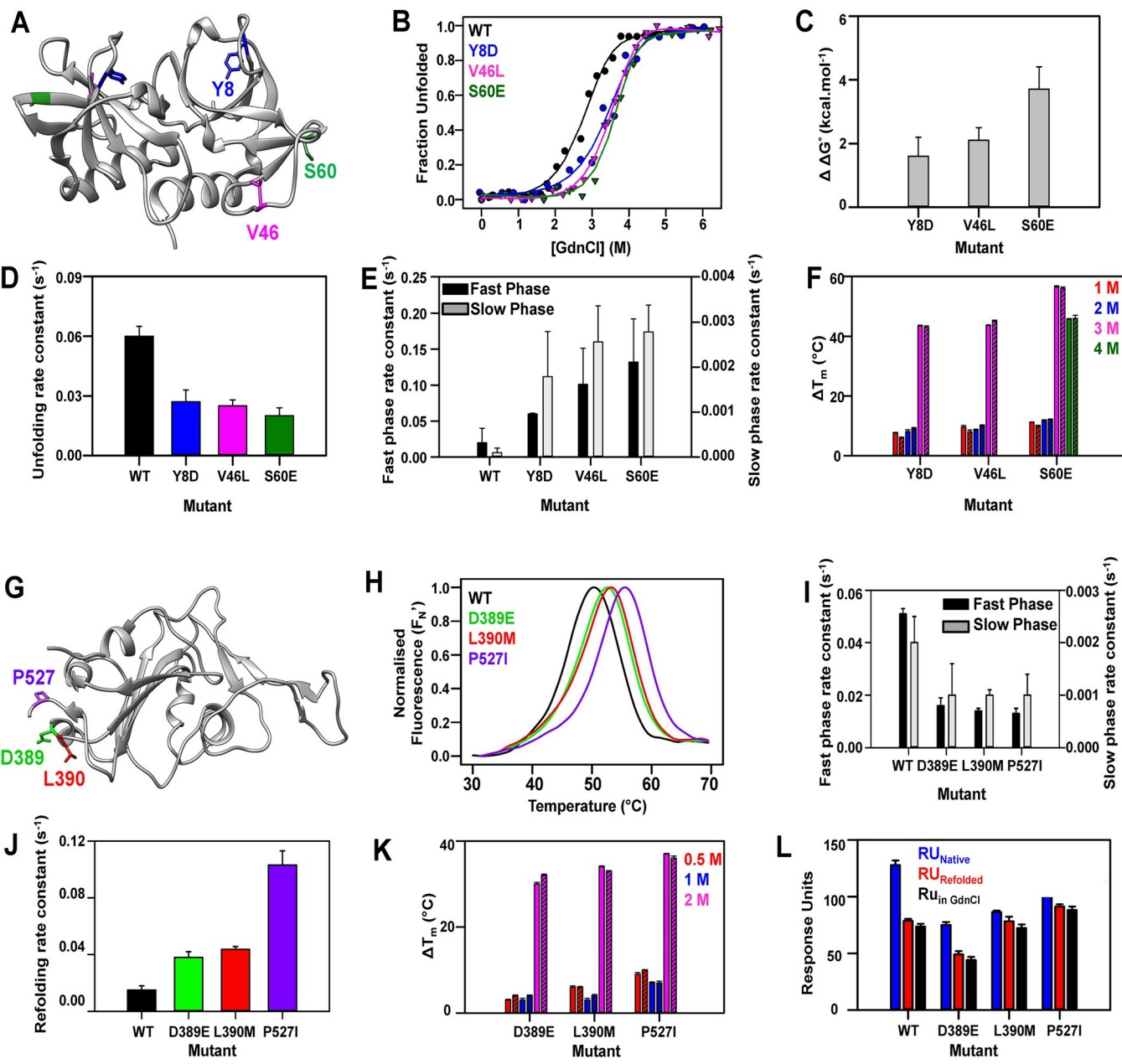

**Fig 5. Enhancement in protein stability by highly stable suppressor mutations in CcdB and mRBD. (A-F) CcdB:** (A) The distal suppressors Y8D, V46L, S60E located on an exposed region mapped onto the crystal structure of CcdB (PDB ID 3VUB [71]). (B) Equilibrium GdnCl denaturation profiles assayed by nanoDSF. (C) Differences in thermodynamic stability assayed by chemical denaturation, $\Delta\Delta G°$ $(\Delta\Delta G° = \Delta G°_{Mutant} - \Delta G°_{WT})$ (in grey) of the different CcdB suppressor mutants are plotted. (D-E) The observed rate constants of unfolding (4.5 M GdnCl) and the observed rate constants of the fast phase (black) and slow phase (grey) of refolding (2 M GdnCl) of different mutants. (F) The difference in thermal stability, $(\Delta T_m = T_{m_{Mutant}} - T_{m_{WT}})$ of 5 μM of native proteins in 1 M, 2M, 3M, 4M GdnCl (solid bars), and refolded CcdB proteins in the same concentrations of GdnCl (striped bars). **(G-L) mRBD:** (G) The distal suppressors D389E, L390M, P527I located on an exposed region mapped onto the crystal structure of RBD (PDB ID 6ZER [78]). (H) Thermal unfolding profiles of 10 μM of purified WT and mRBD mutants. (I-J) The observed rate constants of fast phase (black) and slow phase (grey) of unfolding (3 M GdnCl) and the observed rate constants of refolding (0.5 M GdnCl) of different mutants. (K) The difference in thermal stability, $(\Delta T_m = T_{m_{Mutant}} - T_{m_{WT}})$ of 5 μM of native mRBD proteins in 0.5 M (red), 1M (blue), 2M (pink) GdnCl (solid bars), and refolded mRBD proteins in the same concentrations of GdnCl (striped bars). (L) Binding of 50 nM of native mRBD proteins (blue), native mRBD proteins in 0.5 M GdnCl (red solid) and refolded mRBD proteins in 0.5 M GdnCl (red striped) with ACE2-hFc neutralizing antibody is shown. The error bars wherever shown represent the standard deviation from two independent experiments, each performed in duplicates.

Table), whereas the refolding rates of the suppressors were 2.5–7 times faster than the WT (Figs 5J and S6C and S8 Table). Further, the proteins were refolded in 0.5, 1, and 2 M GdnCl and subjected to thermal denaturation with native proteins in the same GdnCl concentrations as control (Fig 5K). The WT could refold till 1 M GdnCl, whereas the suppressors D389E, L390M and P527I refolded back to the native state even at 2 M GdnCl (S6I–S6K Figs). The binding of the native proteins, native proteins in 0.5 M GdnCl and refolded proteins in 0.5 M GdnCl with ACE2-hFc neutralizing antibody were also measured using ProteOn (Fig 5L). All the refolded proteins showed binding to the ACE2-hFc, indicating that the proteins were properly refolded back to their functional conformation (S6L–S6O Fig) and that chemical denaturation of RBD was reversible.

## Structural insights into stabilization by CcdB suppressor mutants

The structures of the S12G, V46L and S60E mutants of CcdB were solved to resolutions of 1.63 Å, 1.35 Å and 1.93 Å respectively. (Fig 6).

The structures of S12G, V46L and S60E (Fig 6A, 6D and 6G) consist of a single chain in the asymmetric unit, with two chloride ions. One of the ions, which is also present in the WT structure 3VUB, interacts electrostatically with H85[Nε2], R86[NH1], H55[N] and is involved in a hydrogen bonding interaction with a symmetry equivalent T7[Oγ1]. The second Cl⁻ ion in S12G interacts with S38[N], R15[NH2] and a water O[220]. Although there is a water molecule at this position in the WT structure (3VUB), addition of a water molecule in S12G results in an unusually low B-factor whereas a Cl⁻ ion fits well without any negative density and a B-factor of 20 Å². The density for the last residue I101 was not visible in the map for S12G. The electron density map in the region of residues 40–45 for S12G, 43–45 for V46L and 41–42 for S60E was very poor, as a result the side chains could not be fitted. One of the residues in S12G, R40 lies outside but close to the allowed region of the Ramachandran Plot. The mutant structures are very similar to the WT structure (3VUB) with an RMSD of 0.26, 0.39 and 0.39 Å for S12G (Fig 6B), V46L (Fig 6E) and S60E (Fig 6H) respectively.

For S12G, the variations are mainly in the loop regions between Y8-Y14 and A39-V46, indicated in Fig 6B by stars. There are two water molecules (254 and 228) in S12G in place of the two conformers of S12[OH] of 3VUB (Fig 6C). A cluster of water molecules at a hydrogen bonding distance from G12 stabilizes the loop and anchors it via interactions with the backbone atoms of neighbouring residues (Fig 6C) reducing the average B-factor in this region (Fig 6J). These two water molecules substitute for the hydroxyl group of both the conformers of serine in the WT structure. Since the S12G has an additional Cl⁻ ion, an additional comparison was done with the structure 4VUB (WT CcdB) which has the second Cl⁻ ion in the same position as found in S12G. It was found that although the absolute B-factor of S12G and 4VUB were similar in the 8–12 region and lower than that of 3VUB, it was lowest for S12G in the region 39–46 amongst the three structures. The relative B-factors are very similar in both 3VUB and 4VUB when normalized, so the 3VUB structure was used as a reference.

For V46L, the loop region A39-V46, exhibits a major deviation from WT CcdB, as indicated in Fig 6E by a star. L46 is involved in a hydrogen-bond interaction with R62 and hydrophobic interactions with M64 (Fig 6F). The hydrogen-bond interactions are formed between the main chain oxygen atom of L46 and side chain nitrogen atoms of R62 (NH1, NH2). The average B-factors of the V46L structure are lower than the WT with the most reductions in the loops 8–14 and 39–46 (Fig 6J).

For S60E also, the major deviation is in the starred loop region A39-V46 (Fig 6H). E60 is involved in a series of salt-bridge interactions with R48, H55 and R62 (Fig 6I). The salt bridge interactions are formed between the side chain oxygen atoms of E60 (OE1 and OE2) and side

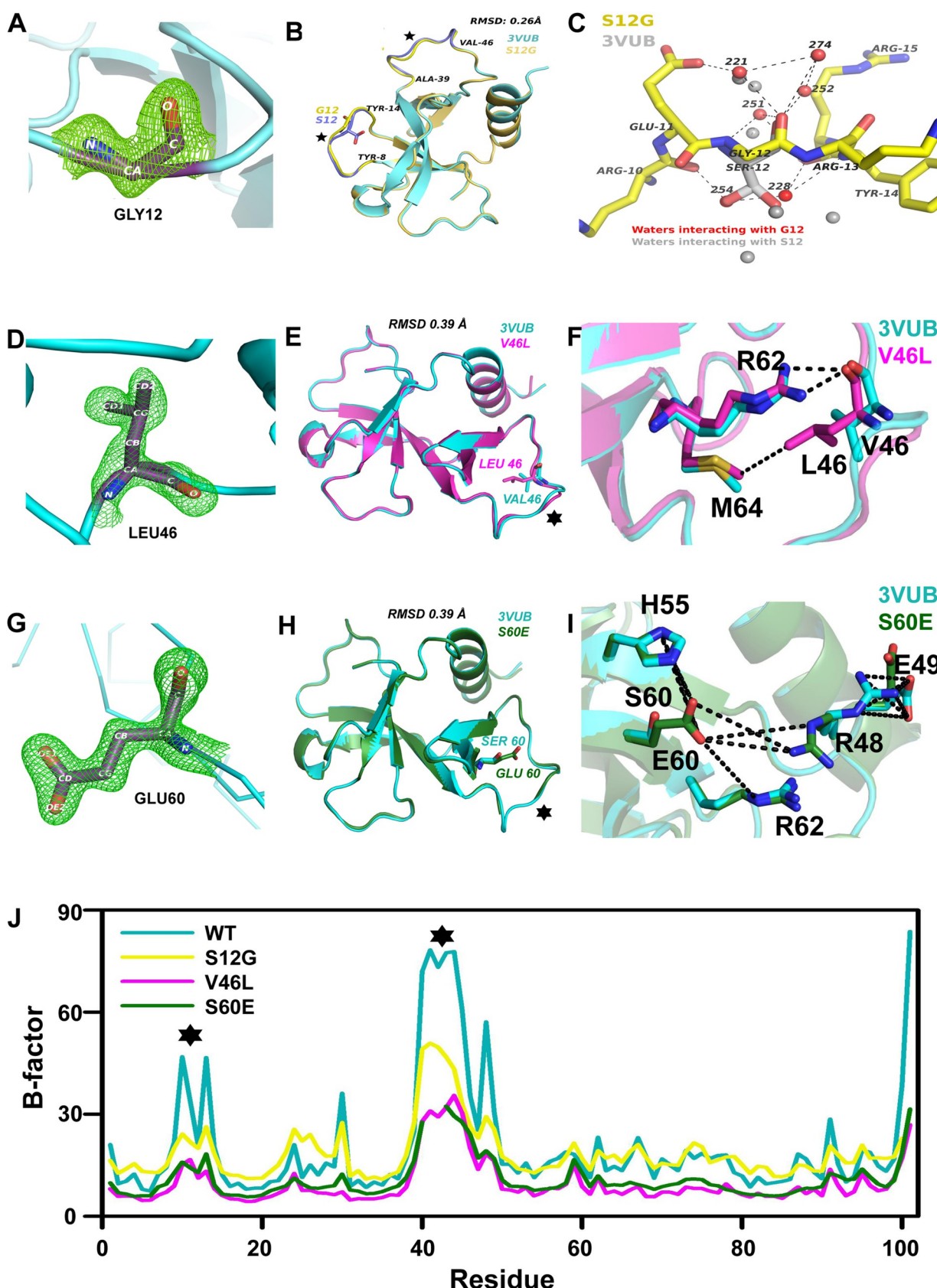

**Fig 6. Structures of stabilised CcdB mutants.** (A) Composite omit map at residue 12. (B) Structural superposition of WT (3VUB) and S12G monomers, regions displaying deviation are indicated by ★. (C) Network of interactions at the 12th position in S12G. WT structure is shown with a grey backbone. S12 in 3VUB adopts two conformations with partial occupancy, the position of the corresponding hydroxyl group in each conformation is taken up by two water molecules in S12G. Water molecules directly interacting with G12 are shown in red and the corresponding water molecules in 3VUB in grey. (D) Composite omit map at residue 46. (E) Structural superposition of WT and V46L monomers, regions displaying deviation are indicated by ★. (F) Network of interactions at the 46th position in V46L. WT structure is shown in cyan. Main chain of L46 is involved in H-bond interactions with side chain nitrogen of R62 and the side chain of L46 is involved in hydrophobic interactions with side chain of M64. (G) Composite omit map at residue 60. (H) Structural superposition of WT and S60E monomers, regions displaying deviation are indicated by ★. (I) Network of interactions at the 60th position in S60E. WT structure is shown in cyan. E60 is involved in salt bridge interactions with R48, H55 and R62. (J) Average B-factor plot of the residues in WT, S12G and S60E. Regions with large variability are indicated by "★".

chain nitrogen atoms of R48 (NE, NH1), H55 (ND1, NE1) and R62 (NE). There is a change in the orientation of the mobile R48 side chain resulting in salt bridge interactions with E60 (Fig 6I). The S60E mutation has resulted in reduced B-factor differences between the side chain and main chain in many regions, including R48, resulting in overall stabilisation of the structure. The average B-factors of the S60E structure are also lower than the WT with the most reductions in the loops 8–14 and 39–46 (Fig 6J).

## Enhanced stability is neither necessary nor sufficient for a mutant to act as a global suppressor

While most suppressor mutations described above confer enhanced stability in the WT background, it is not known if all stabilized mutants will act as suppressors. Two such CcdB mutants, L42E and S43T, that enhanced the thermodynamic stability and one mutant M32T that was less thermodynamically stable than WT, similar to E11R were characterised. From our previous studies that have characterised a large number of CcdB mutants by YSD, L42E and S43T were seen to exhibit higher binding than WT and were presumed to be more stable [35]. However, these mutants were not identified as suppressors using YSD [13]. In the same study, M32T was identified as a suppressor [13]. This was surprising as M32 is buried and substitution by a polar residue should be destabilizing. We confirmed this by DSF measurements on purified protein (Fig 7B). We therefore introduced the M32T, L42E and S43T mutations individually in the background of various parent inactive mutants, and the binding to GyrA14 was measured by FACS as described previously [13,35]. It was observed that L42E and S43T failed to enhance the binding to GyrA14 of any of the inactive mutants whereas M32T was able to rescue the folding defect of V20F and L36A inactive mutants (Fig 7A). Next, we characterised the thermodynamic and kinetic stabilities of the purified M32T, L42E and S43T proteins. We observed that though the thermal and chemical stabilities of L42E and S43T were higher than WT (Fig 7B–7D, S9 Table), the folding kinetic parameters were similar to WT (Fig 7E and 7F, S9 Table). Additionally, although M32T was thermodynamically less stable than WT, it showed faster refolding (Fig 7F, S9 Table), thus indicating that faster refolding rather than enhanced stability is sufficient to rescue folding defects of mutants (Fig 7B–7F). We also measured the thermal stability of the refolded proteins in 1.5 M GdnCl and subjected them to thermal denaturation (Fig 7E, S9 Table). The refolded L42E and S43T proteins had higher thermal stabilities while the refolded M32T had lower thermal stability than the refolded WT protein (Fig 7E, S9 Table). These observations demonstrate that enhanced thermal stability alone is insufficient to confer a global suppressor property to a mutant.

## Discussion

In this work, we examine the mechanisms by which a second (suppressor) mutation alleviates the protein defects caused by the initial loss-of-function causing point mutant. Previous

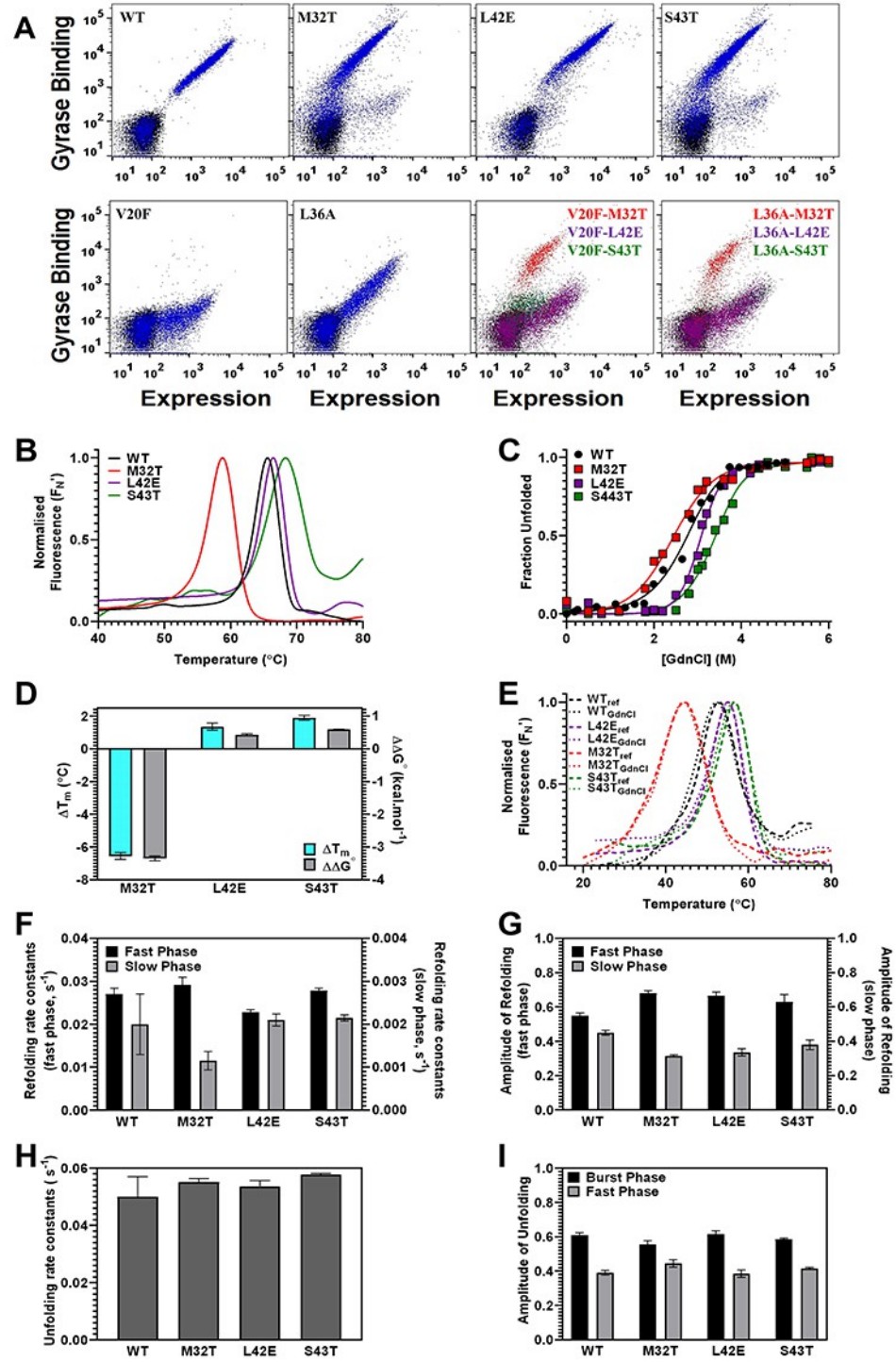

**Fig 7. Enhanced stability is neither necessary nor sufficient for a mutant to act as a global suppressor.** (A) Analysis of yeast cell surface expression and GyrA14 binding of different CcdB mutants and WT. CcdB WT and mutant plots (blue), are overlaid with plot of uninduced cells (black). In the last two panels, V20F-M32T, L36A-M32T plots (red) are overlaid with plots of V20F-L42E, L36A-L42E (purple) and V20F-S43T, L36A-S43T (green), only M32T is able to suppress the deleterious effects of the V20F and L36A mutations. (B-I) Kinetic and thermodynamic characterisation of CcdB mutants. (B) Thermal unfolding profiles of 5 μM of CcdB-WT, M32T, L42E and S43T mutants carried out by nanoDSF. (C) Equilibrium GdnCl denaturation profiles of 5 μM of CcdB-WT, M32T, L42E and S43T mutants carried out by nanoDSF. (D) Difference in thermal $\Delta T_m$ ($\Delta T_m = T_{m_{Mutant}} - T_{m_{WT}}$) (in cyan), and thermodynamic stability assayed by chemical denaturation, $\Delta\Delta G°$ ($\Delta\Delta G° = \Delta G°_{Mutant} - \Delta G°_{WT}$) (in grey) of the CcdB mutants. (E) Thermal

unfolding profiles of 5 μM of native proteins in 1.5 M GdnCl (dotted lines) and refolded CcdB proteins in the same concentration of GdnCl (dashed lines). (F-G) The observed rate constants and amplitudes of the fast phase (black) and slow phase (grey) of refolding (1.5 M GdnCl) of WT and CcdB mutants (see also S9 Table). (H-I) The observed rate constants and amplitudes of unfolding (3.5 M GdnCl) of WT and CcdB mutants. The error bars wherever shown represent the standard deviation from two independent experiments, each performed in duplicates.

studies have shown physically interacting residues to coevolve [36] or mutate with substitutions bearing shape or charge complementarities for stability compensation [1]. The location of suppressor mutations may be either spatially proximal or distal from the site of the original mutation but are usually found on the surface [37]. Global suppressors are expected to have a WT like phenotype, when present as single mutants [1]. Some plausible mechanisms responsible for global suppression are: a) improving the foldability of the protein without impacting stability [6], b) increasing global thermodynamic stability [4,38,39] thereby compensating a folding defect caused by an initial destabilising albeit function altering mutation [7,25,40], c) improving the specific activity of the protein, for example through a mutation at a functionally important residue [5].

In previous studies, we identified several global suppressors of inactivating mutations of the bacterial toxin, CcdB [1,13]. In the present study, we characterised the mechanisms of suppression by the E11R and S12G global suppressors in considerable detail, and extended these studies to four other global suppressors, Y8D, V46L, S60E and M32T. Non active-site, buried mutations typically affect the levels of correctly folded protein [41]. The presence of low levels of active, folded CcdB protein is sufficient to kill the cells and rescue the inactive phenotype caused by the PIMs. This was confirmed *in vivo* by growth assays and estimation of solubility levels. Equilibrium thermal and chemical denaturation studies reveal a large enhancement in the apparent stability of PIM-suppressor proteins with respect to the PIMs in isolation. Surprisingly, the suppressor mutations E11R and M32T show decreased stability, relative to WT CcdB, and the remaining suppressors show marginal improvements in stability. There is however, a non-additivity of the apparent stabilising effect of E11R and S12G in the presence and absence of the PIM. This might be attributed to the fact that the stability of the PIMs are difficult to measure accurately because of their aggregation-prone nature.

Kinetic studies were used to further elucidate the mechanism of action of such distal global suppressors. Several studies have shown the importance of kinetic stability in the evolutionary optimization and regulation of protein function [8,42–44]. Kinetic destabilisation leads to diseases associated with protein misfolding [45,46]. Therefore, a mutation which enhances kinetic stability can be important in both physiological and biopharmaceutical contexts, for example increasing shelf life of monoclonal antibodies and enhancing vaccine immunogenicity [10,47].

Relative to the WT protein, we observe that the increment in the folding rate parameters is typically larger than the decrement in the unfolding rate parameters. This suggests that additional favourable interactions resulting from the mutation are formed prior to the folding transition state, lowering its energy, relative to the unfolded state. In the case of the CcdB S12G, V46L and S60E; β-lactamase M182T and p53 N329Y mutants, for which crystal structures are available, additional non-covalent interactions present, relative to corresponding WT structure are seen. For these CcdB suppressors, the structural changes are complex and unlikely to be predicted from modelling studies.

In order to understand the role of thermodynamic versus kinetic stability as a criterion for global suppression, we investigated several other CcdB mutants based on their relative thermal stability and ability to act as a suppressor. Similar to E11R, the M32T mutant was also thermodynamically less stable as compared to the WT, but was still able to act as a suppressor (Fig 7). In contrast, L42E and S43T, though more stable than the WT, were unable to suppress the

folding defects of the inactive mutants, suggesting that enhanced thermodynamic stability alone is not essential for suppression. Therefore, the findings collectively indicate the role of kinetic stability, in particular an increase in the refolding rate constant as being primarily responsible for the suppressor phenotype.

For all the mutant variants of all the proteins used in the study, we observe that the native state of a suppressor mutant is not destabilised relative to its wild type counterpart, except for the functionally active CcdB E11R, a charge reversal substitution, and M32T which is a polar substitution at a dimer interface, resulting in mild thermodynamic destabilisation relative to wild type (S1, S2 and S9 Tables). Previous studies have reported that the M182T suppressor mutant in TEM-1 β-lactamase stabilises the enzyme by recruiting newly formed hydrogen bonds mediated by threonine182 and adjacent water molecules [21]. Previous reports have also shown that core-engineered mutations like M69I and L76N destabilise the native state of the lactamase enzyme [6], similar to the low thermodynamic stabilities observed for these inactive mutants in our thermal and chemical denaturation assays (S5 Table). Addition of M182T suppressor mutation to these inactive mutants stabilises their folded states and restores their functional defects (S5 Table).

Similarly, for the DNA-binding domain of p53, V143A and V157F are core mutations that destabilise the native state of the protein and lead to low expression of soluble functional protein levels (S6 Table). V143A causes perturbations in almost all the residues of the β-sandwich and DNA-binding surface [48]. V157F, one of the strongest destabilising oncogenic mutant, causes side-chain rearrangements in the core of the protein [33]. N239Y, suppressor mutant, stabilises the native state of the protein by introducing new hydrophobic contacts and hydrogen-bonds with water molecules, which were absent in the wild type protein [33]. It can, therefore, be hypothesised that both the destabilised mutants, V143A and V157F, could be functionally rescued and stabilised by the suppressor by stabilisation of their folded states upon introduction of alternative favourable interactions by Y239 in the protein.

On similar lines, in this study, the crystal structures of CcdB mutants reveal that the suppressors S12G, V46L and S60E possess novel hydrogen-bonds with water molecules or adjacent residues, that were absent in the wild type protein and are likely responsible for stabilisation of the native state of the protein by these suppressors, as described above.

None of the CcdB suppressor mutations are seen in naturally occurring paralogs. The intrinsically disordered C-terminal domain of the cognate antitoxin CcdA, facilitates the rejuvenation of the poisoned Gyrase-CcdB complex by forming a transient ternary complex prior to extracting CcdB from its complex with Gyrase. Most of the identified suppressors are present in the distal loops 8–15 and 39–52 which are directly involved in CcdA binding. The CcdB positions 10, 12 and 46 are involved in CcdA binding and mutations at these positions as well as the nearby E11 residue will affect CcdA interaction and conformation of the CcdA interacting loop from residues 8 to 15 ([8]YKRESRYR[15]) (S7 Fig). S60E, though not involved directly in CcdA interaction, alters the conformation and rigidity of the 39–46 loop that contacts CcdA and might therefore affect CcdB function (S7 Fig). V46L also similarly affects the rigidity of the 8–14 and 39–46 loops important for CcdA binding and GyrA14 rejuvenation [49,50]. In CcdB, the E11R and M32T suppressor mutations had lower stability than WT and conversely, the L42E mutant which is more stable than WT, failed to act as a suppressor. These data demonstrate that while most suppressor mutations show small stabilization effects in the WT background, increased stability is neither necessary nor sufficient for global suppressor mutations. In separate studies from our laboratory, it was observed that though the individual suppressor mutations do not greatly alter affinity towards CcdA, mutations at all these positions significantly affected the rejuvenation process [50]. The functional importance of these CcdB residues, explains why the experimentally identified global suppressor mutations are not found in

naturally occurring ccdB genes. More detailed assessments of differences in hydrogen-bonding, possible intramolecular interactions between proximal loops and solvent accessibility are shown in S9 Fig and S13 and S14 Tables.

The suppressors are largely identified in distal loops, far from the inactive mutations which are present in the core. There are various possible explanations. Firstly, the core is well packed, and it may not be easy to find mutations that improve upon this in the context of the wild type protein. In the context of a core PIM, destabilizing effects can be alleviated by proximal suppressors which restore packing/H-bonding, however as discussed previously [1], these will be allele specific, not global suppressors. In contrast to the core, there are fewer constraints for surface residues, especially in flexible loops, hence stabilizing mutations are enriched in such sites. In the specific case of CcdB, the suppressor mutations result in new native-state interactions that stabilize the loop and/or surrounding loops as observed in the current study. This results in overall stabilisation of the protein. Previous studies have also demonstrated that the deletion of a 44–49 omega loop in both wildtype staphylococcal nuclease (E43SNase) and a mutant E43D nuclease (D43SNase), resulted in both increased activity and stability as compared to their respective parent enzymes [51,52].

In the present work, we focus on protein folding kinetics studied *in vitro* whereas *in vivo*, for several proteins, folding occurs co-translationally and/or is assisted by molecular chaperones [53]. How suppressor mutations affect the kinetics and yield of co-translational or chaperone mediated folding/unfolding is beyond the scope of the present work and it is likely that the different proteins studied make use of different chaperone systems and span both cytoplasmic and periplasmic folding compartments. Nevertheless, it is clear from the present studies that acceleration in folding kinetics *in vitro* is associated with enhanced yield of active protein *in vivo* whether in the context of *E. coli* or yeast [13,35]. A previous study from our laboratory, which looked at the effects of overexpression of a number of different chaperones on ccdB mutant phenotypes reported the rescue of inactive, folding defective CcdB mutants which occurred exclusively in the two *E. coli* strains overexpressing ATP-independent chaperones (SecB and Trigger factor), that act early in *in vivo* folding and not in the strains overexpressing ATP-dependent chaperones that act in the later stages of the folding pathway [8]. One of the important inferences from this study was that mutational effects on folding, rather than stability, influenced CcdB mutant activity *in vivo*, consistent with the present results. To date, there have been reports on the detailed investigation of the Hsp70 and Hsp90-mediated stability and activity of p53-DBD [54–56], the GroEL/ES chaperonin system making transient interactions and inhibiting the folding of β-lactamase precursor [57,58], and of a strongly bound complex of the GroEL chaperone with the receptor-binding domain of the SARS CoV2 spike protein [59]. Furthermore, the effect of overexpression of GroEL/ES chaperonin system and the deletion of Lon protease on the fitness of a stabilized mutant L28R in the background of WT and two destabilized mutants P21L, A26T was investigated for the trimethoprim (TMP) resistance of *Escherichia coli* dihydrofolate reductase (DHFR), where it was observed that the levels of GroEL/ES chaperonins and Lon protease affect the intracellular steady-state concentration of DHFR in a mutation-specific manner and there are complex, epistatic interactions between the three mutations [60]. However, none of the above discuss the effects of suppressor mutations on chaperone mediated folding kinetics.

We have summarized the effect of the suppressor mutations characterized in the present study on various thermodynamic and kinetic parameters in the background of both WT and inactive mutants (Fig 8, S11 Table). For the PIM/Suppressor pair analysis, we excluded the p53-DBD double mutants since we did not have the corresponding inactive mutants for comparison. Relative to the WT, the individual suppressors have marginal enhancement in thermal stability ($\Delta T_m$) and significant changes in the chemical stability ($\Delta\Delta G°$) (Fig 8A). However, in

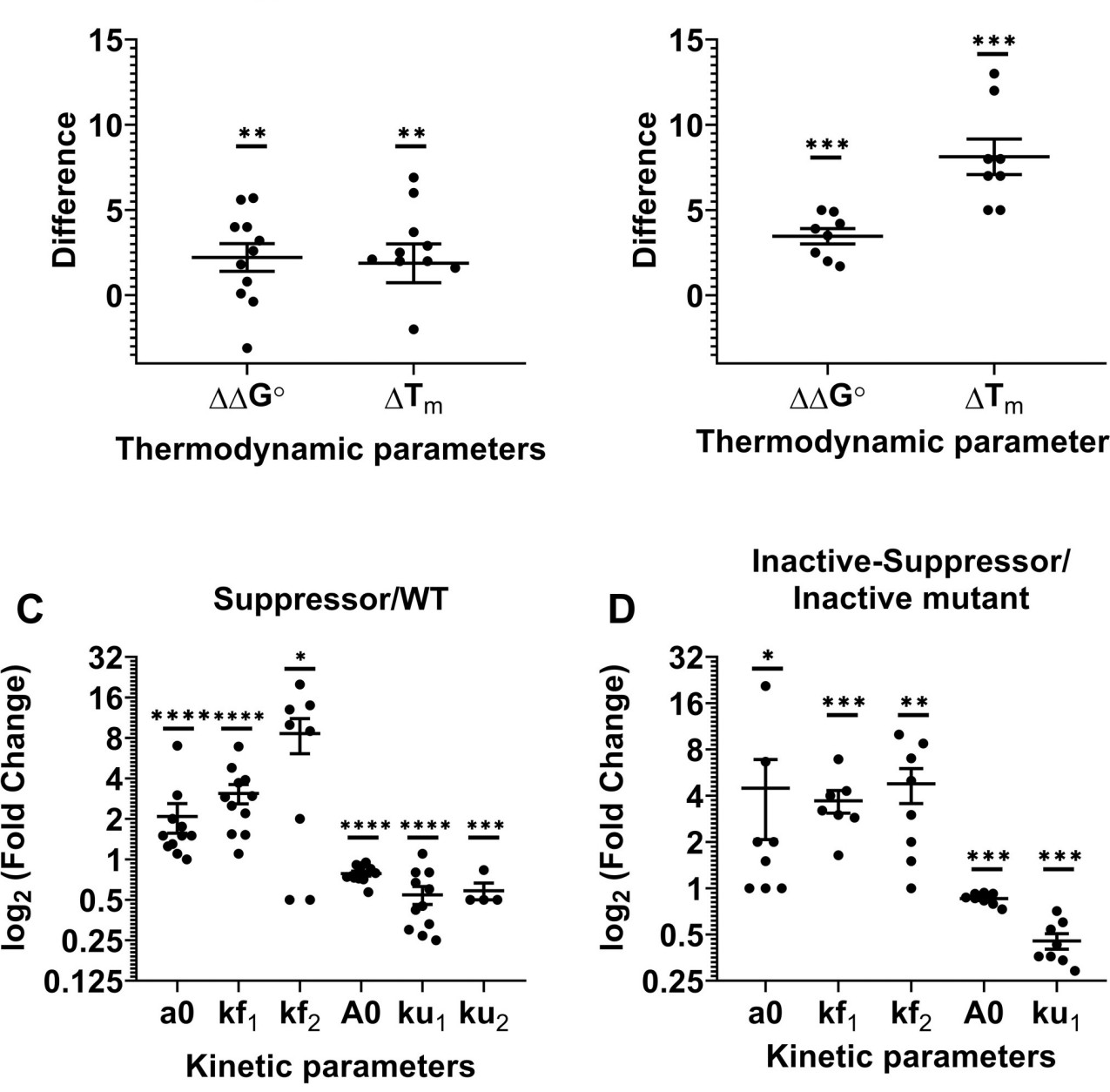

**Fig 8. Suppressor mutations have larger effects on refolding kinetics than on unfolding kinetics or protein stability parameters.** (A-B) Distribution of $\Delta T_m$ (°C) and $\Delta\Delta G°$ (kcal/mol) (Mean±SEM) for the suppressor mutations in the background of WT (A) or inactive mutants (B). Mann Whitney non-parametric test was performed for each of these parameters to examine if they are significantly different from zero. (C-D) $Log_2$ fold change of various kinetic parameters (Mean±SEM) for the suppressor mutations in the background of WT (C) or inactive mutant (D). Mann Whitney non-parametric test was performed for each of these parameters. The mean of the distributions of the values for each of the parameters are significantly higher than $log_2(2)$ for refolding, and lower than $log_2(2)$ for unfolding. *, ** and *** indicate values of P < 0.05, < 0.005 and < 0.0005 respectively. $\Delta\Delta G°$ and $\Delta T_m$ represent the difference in Gibbs free energy and apparent melting temperature $T_m$ for suppressor containing protein relative to the corresponding values for either WT (A) or Parent Inactive Mutant (B). a0, $kf_1$, $kf_2$, A0, $ku_1$, $ku_2$ are the burst phase amplitude for refolding, rate constant of fast phase of refolding, rate constant of slow phase of refolding, amplitude of burst phase of unfolding, rate constant of fast phase of unfolding, and the rate constant of slow phase of unfolding respectively.

the background of the inactive mutants, the suppressors cause significant enhancement in both thermal ($\Delta T_m$) and chemical stabilities ($\Delta\Delta G°$) (Fig 8B). One suppressor has a similar rate constant as the WT for the slow phase of refolding and one PIM-suppressor pair has a similar burst phase of refolding as that of the inactive mutant. However, all other suppressors or PIM-suppressor pairs have altered refolding and unfolding kinetics. To further delineate the parameters which are most effected by the suppressor mutations, we calculate the average fold change of each parameter by the suppressor mutations (S11 Table) as follows:

$$P_{avg} = \frac{1}{n}\left(\sum_1^n P_i\right), \qquad\qquad 1$$

where $P_{avg}$ is the average fold change of a parameter P,

$P_i$ is the fold change of that parameter in the background of the suppressor which is $P_i = \frac{P_{suppressor}}{P_{wt}}$ or $P_i = \frac{P_{inactive,suppressor}}{P_{inactive}}$

and n is the number of mutants. A non-parametric Mann Whitney test is performed to show that the mean of the distributions of each of these values of $P_i$ is significantly higher (or lower) than 1 (Fig 8C and 8D, S11 Table) and confirm that enhancement of refolding kinetics by suppressor mutants is statistically significant in the background of both WT and inactive mutations. The data clearly demonstrate that apparent effects of the suppressor mutation are larger in the context of the PIM than in the WT protein. In addition, effects of the suppressor mutation on refolding rate parameters (both burst phase amplitude as well as refolding rate constants) are larger than corresponding changes in unfolding kinetic parameters. Thus, all aspects of refolding are affected, albeit to different extents in different proteins. Destabilizing mutations at buried sites typically slow down the folding process *in vitro*, we speculate that this also holds true *in vivo*. In the crowded environment of the cell, this might facilitate off-pathway aggregation. Suppressor mutations that accelerate refolding can counteract this, enhancing the yield of properly folded, functional protein *in vivo*.

The different parameters of the folding process were individually analysed for all the protein systems used in this study, except for p53 since only one suppressor substitution N239Y was used in the study, and the p53-DBD double mutants could not be used for analysis since we did not have the corresponding inactive mutants for comparison. Upon analysing the nature of refolding transitions, it is clear that proteins belonging to varied classes show different refolding kinetics, wherein mRBD (WT and mutants) shows monophasic refolding, whereas mutants and WT proteins of CcdB, DNA binding domain of p53 and TEM-1 β-lactamase show biphasic refolding. We find for CcdB suppressors, the slow phase of refolding has a significant contribution in rescuing the folding defect and increasing the stability whereas for TEM-1 β-lactamase and mRBD mutants, we find that the suppressors rescue the folding defect by increasing the refolding rate constant of the fast phase (S8 Fig, S12 Table).

For relatively stable proteins such as CcdB ($T_m$ = 66°C) and mRBD ($T_m$ = 50.4°C) which are not folding defective or aggregation prone, it is not easy to isolate mutants with improved stability, or to screen for suppressors. This can be overcome by first introducing a destabilizing mutation (Parent Inactive Mutation or PIM) into a saturation mutagenesis library, followed by screening for suppressors [1]. We have recently shown that such an approach [10,13] can robustly be used to identify multiple individual suppressor mutations. While each suppressor significantly improves the activity of the PIM, as seen in the present work, these typically have only a small stabilization effect when introduced in the WT background i.e. the apparent stabilization of PIM by suppressor is not quantitatively transferable to WT. Overall, the effects of suppressors on the refolding rate parameters were larger than on the unfolding rate

parameters. This was observed across multiple suppressors in multiple proteins, suggesting this to be the primary mechanism through which such suppressors function.

## Materials and methods

### Plasmids and host strains

**CcdB.** The ccdB gene was cloned under the control of $P_{BAD}$ promoter in pBAD24 vector [14]. Two *Escherichia coli* host strains *Top10*pJAT and *Top10GyrA* were used. *Top10*pJAT is a CcdB sensitive strain and was used for screening the phenotypes. The pJAT8araE plasmid which encodes for the arabinose transporter AraE was introduced into the *TOP10* strains to ensure that in all cells there is approximately equal amounts of arabinose uptake [61]. *Top10GyrA* is resistant to the action of CcdB toxin and was used for monitoring the expression of mutant proteins. The strain contains a GyrA462 mutation in its genome that prevents CcdB from binding to Gyrase [9]. The *Saccharomyces cerevisiae* strain EBY100 was used for yeast surface display to monitor the binding and expression of the displayed proteins cloned in the yeast surface display vector pPNLS [62].

**TEM-1 and p53-DBD.** WT and mutant TEM-1 β-lactamase with a C-terminal 6xHistidine tag were cloned and expressed under the control of the T7 promoter in the pET-24a vector. The native signal sequence was used for efficient secretion in the *Escherichia coli* host strain BL21 (DE3) pLysE. WT and mutant p53-DBD genes with N-terminal 6xHistidine tag were cloned and expressed under the control of the T7 promoter in the pET-15b vector. *Escherichia coli* host strain BL21 Rosetta (DE3) was used for expressing the p53-DBD proteins.

**mRBD.** mRBD WT and mutants were expressed from mammalian cell culture as described previously [34] under the control of the CMV promoter along with a tPA signal sequence for efficient secretion.

### Mutagenesis

**CcdB.** For single mutants V5F, Y8D, E11R, S12G, V18W, V20F, M32T, L36A, S43T, V46L, S60E and L83S, as well as for double mutants, V18W-E11R, V20F-E11R, L36A-E11R, L83S-E11R, V20F-M32T, L36A-M32T, V18W-L42E, V20F-L42E, L36A-L42E, L83S-L42E, V20F-S43T and L36A-S43T, the ccdB gene was amplified in two fragments with the desired point mutations. The fragments had overlapping regions (introduced during PCR) of 15–20 nucleotides, which were then recombined using Gibson assembly or *in vivo* recombined with pPNLS vector for YSD as described earlier [63]. Amplification was done using Phusion Polymerase from NEB as per the manufacturer's protocol. The double mutants V5F-S12G, V18W-S12G, V20F-S12G, L36A-S12G and L83S-S12G were synthesized by GeneArt (Germany).

**TEM-1 and p53-DBD.** The TEM-1 β-lactamase WT and mutants M182T, M69I, M69I-M182T, L76N, L76N-M182T and p53-DBD WT and mutants N239Y, V143A, V157F, V143A-N239Y, V157F-N239Y were synthesized by GenScript (USA).

**mRBD (331–532).** The mRBD (331–532) codon optimised for human cell expression was synthesized by GenScript (USA) [34]. For single mutants D389E, L390M and P527I, the RBD gene was amplified in two fragments with the desired point mutations. The fragments had overlapping regions (introduced during PCR) of 15–20 nucleotides, which were then recombined using Gibson assembly. Amplification was done using Phusion Polymerase from NEB as per the manufacturer's protocol.

### Protein expression and purification

**CcdB.** WT CcdB and all mutants were expressed from the arabinose promoter $P_{BAD}$ in the pBAD24 vector in the CcdB resistant *Top10GyrA* strain of *E. coli*. The purification of the

CcdB mutants were carried out as described previously [16]. Briefly, 500 mL of LB medium (HiMedia) was inoculated with 1% of the primary inoculum and grown at 37˚C until the $OD_{600}$ reached 0.6. Cells were then induced with 0.2% (w/v) arabinose and grown at 37˚C for 5 hours for WT CcdB, Y8D, E11R, S12G, M32T, L42E, S43T, V46L, S60E and the double mutants V18W-S12G, V20F-S12G, L36A-S12G and L83S-S12G, at 25˚C overnight for the inactive mutants V18W, V20F, L36A and L83S and at 20˚C overnight for the double mutants L36A-E11R and L83S-E11R. Cells were harvested, re-suspended in HEG re-suspension buffer pH 7.4 (10 mM HEPES, 50 mM EDTA, 10% glycerol containing 10 mM PMSF) and lysed by sonication. The supernatant was incubated with Affi-gel15 (Biorad) coupled to CcdA peptide (residues 46–72) and incubated overnight at 4˚C. The unbound fraction was removed and washed with five times the bed volume of coupling buffer pH 8.3 (0.05 M Sodium Bicarbonate, 0.5 M Sodium Chloride). The elutions were carried out with 0.2 M Glycine, pH 2.5 into a tube containing an equal volume of 400 mM HEPES, pH 8.4, 4˚C [8,16]. The eluted fractions were subjected to 15% Tricine SDS-PAGE and the protein concentration was determined. Yield for all mutants varied from 0.3–12 mg/L depending upon the amount of protein in the soluble fraction. Fractions containing pure protein were pooled and stored at −80˚C.

V5F and V5F-S12G could not be purified using affinity chromatography against immobilised CcdA because of their low expression, solubility and inability to bind to the CcdA column. V18W-E11R and V20F-E11R could not be used for further biophysical studies owing to their high tendency to aggregate.

**TEM-1 β-lactamase.** The TEM-1 β-lactamase mutants were purified as described previously [22] with slight modifications. The recombinant BL-21 (λDE3, plysE) strains were grown in TB medium at 37˚C containing 50 μg/mL kanamycin until the $OD_{600}$ reached 0.8 and protein expression was induced by addition of 1.0 mM IPTG. The induced cultures were grown overnight with shaking at 30˚C and were harvested by centrifugation. The periplasmic fraction was obtained by osmotic shock by resuspending first in lysis buffer, pH 7.0 (10 mM HEPES, 0.5 mM EDTA, 20% sucrose, 0.05% SDS, lysozyme and protease inhibitor) at 37˚C with shaking for 1 hr, followed by addition of an equal volume of ice cold milliQ water, incubated at 4˚C for an hour, followed by addition of 100 μL of 2 M $MgCl_2$. The clarified lysates were loaded on 2 ml of Q-Sepharose fast flow (Amersham Biosciences, Uppsala, Sweden). The elutes obtained by a gradient of 100-500mM NaCl in 10 mM HEPES, 10% glycerol pH 7.0, were further subjected to Ni-NTA purification by mixing with 2 mL of Ni-Sepharose resin (GE Healthcare) for 4 hrs and the bound proteins were eluted with a gradient of 100–500 mM imidazole in 10 mM HEPES, 300 mM NaCl, 10% glycerol (pH 7.0). The purified proteins were subjected to 15% Tricine SDS-PAGE, concentrated, buffer exchanged to remove imidazole, and finally stored in storage buffer (10 mM HEPES, 300 mM NaCl, 10% glycerol, pH 7.0) at -80˚C until further use. Yield for all mutants varied from 0.5–10 mg/L.

**p53-DBD.** The p53-DBD mutants were purified as described previously [64] with slight modifications. The recombinant BL-21 Rosetta (DE3) strains were grown in TB medium at 37˚C containing 100 μg/mL ampicillin until the $OD_{600}$ reached 1.0. 100 μM $ZnSO_4$ was added and incubated at 25˚C for 30 mins. Protein expression was induced by addition of 1.0 mM IPTG. The induced cultures were grown for 20 hours with shaking at 25˚C and were harvested by centrifugation. Cells were harvested, re-suspended in lysis buffer pH 7.2 (50 mM $NaH_2PO_4$, 100 mM NaCl, containing 10 mM PMSF and 10 mM DTT) and lysed by sonication. The clarified lysates were loaded first with 2 ml of DEAE Sepharose fast flow (GE Healthcare) and the elutes obtained by a gradient of 100–500 mM NaCl in 50 mM $NaH_2PO_4$, pH 7.2, were further subjected to Ni-NTA purification by mixing with 2 mL of Ni-Sepharose resin (GE Healthcare) for 4 hrs. The bound proteins were eluted with a gradient of 100–500 mM imidazole in 50 mM $NaH_2PO_4$, 500 mM NaCl, (pH 7.2) containing 10 mM DTT. The purified proteins were

subjected to 15% Tricine SDS-PAGE, concentrated and finally stored in storage buffer (50 mM NaH$_2$PO$_4$, 500 mM NaCl, 500 mM imidazole, 10 mM DTT, pH 7.2) at -80°C until further use. The inactive mutants V143A, V157F, could not be used for biophysical studies because of their low expression, solubility, poor yields and high tendencies towards aggregation after purification. Yield for the WT and the mutants varied from 0.5–2 mg/L.

**mRBD (331–532).** Expression and purification of the mRBD and the single mutants D389E, L390M and P527I was carried out as described previously [34]. Briefly, proteins were expressed by transient transfection of Expi293 cells and purified by Ni-NTA chromatography. The eluted fractions were pooled and dialysed thrice using a 3–5 kDa (MWCO) dialysis membrane (40mm flat width) (Spectrum Labs) against 1X PBS, pH 7.4 (storage buffer). The eluted fractions were subjected to 15% Tricine SDS-PAGE and the protein concentration was determine by measuring the A$_{280}$ and using an extinction coefficient of 33850 M$^{-1}$cm$^{-1}$.

## *In vivo* activity of the different CcdB mutants

WT and mutant CcdB were transformed into *E. coli Top10*pJAT, grown for 1 hour in 1 ml LB media containing 0.2% glucose (highest repressor level to avoid leaky expression). After 1 hour, the cells were pelleted and glucose was removed by subjecting cells to three washes with 1 ml LB. Finally, equal amounts of cells resuspended in 1 ml of LB media and serially diluted were spotted on seven agar plates (LB agar plates containing 100 μg/mL ampicillin, 20 μg/mL gentamycin) containing various amounts (%) of glucose (repressor) and arabinose (inducer) concentrations (i.e. $2\times10^{-1}$% glucose, $4\times10^{-2}$% glucose, $7\times10^{-3}$% glucose, 0% glucose/arabinose, $2\times10^{-5}$% arabinose, $7\times10^{-5}$% arabinose and $2\times10^{-2}$% arabinose) at and grown 37°C. Since active CcdB protein kills the cells, colonies are obtained only for mutants that show an inactive phenotype under the above conditions [8].

## *In vivo* solubility estimation

**CcdB.** Solubility levels were monitored for all the single and double mutants in *E. coli Top10GyrA* in the presence of 0.2% arabinose as described previously [8]. Cultures were grown in LB media, induced with 0.2% arabinose at an OD$_{600}$ of 0.6 and grown for 5 hours at 37°C. $2\times10^9$ cells were centrifuged (1800g, 10 min, RT). The pellet was resuspended in 500 μL HEG buffer pH 7.4 (10 mM HEPES, 50 mM EDTA, 10% glycerol containing 10 mM PMSF) and sonicated. 250 μL was taken as Total Cell Lysate and the remaining 250 μL was centrifuged (11000 g, 10 min, 4°C). 250 μL was taken as supernatant and the pellet was resuspended in 250 μL HEG buffer pH 8.4 and all the fractions were subjected to 15% Tricine SDS-PAGE. Solubility of all mutants was quantitated by estimating relative amounts of CcdB in supernatant and pellet fractions by the Geldoc software (Quantity One) [8,65].

**p53-DBD.** Solubility levels were monitored for all the single and double mutants in *E. coli* BL-21 Rosetta (DE3) cells. Cultures were grown in LB media till they attain OD$_{600}$ values of 1.0, followed by incubation with 100 μM ZnSO$_4$ at 25°C for 30 minutes and induction with 1.0 mM IPTG, for 20 hours at 25°C. $2\times10^9$ cells were centrifuged (1800g, 10 min, RT) and the pellet was resuspended in 500 μL lysis buffer pH 7.2 (50 mM NaH$_2$PO$_4$, 100 mM NaCl, containing 1 mM PMSF and 10 mM DTT) and sonicated. 250 μL was taken as Total Cell Lysate and the remaining 250 μL was centrifuged (11000 g, 10 min, 4°C). 250 μL was taken as supernatant and the pellet was resuspended in 250 μL lysis buffer pH 7.2 and all the fractions were subjected to 15% Tricine SDS-PAGE. Solubility of all mutants was quantitated by estimating relative amounts of the p53 DBD protein in supernatant and pellet fractions by the Geldoc software (Quantity One) [8,65].

## Thermal stability measurements by TSA and nanoDSF

**CcdB.** The thermal shift assay was conducted in an iCycle Q5 Real Time Detection System (Bio-Rad, Hercules, CA) using 4 μM of the CcdB protein in 200 mM HEPES, pH 8.4 and 2.5X Sypro orange dye in a 96-well iCycler iQ PCR plate. The plate was heated from 20 to 90˚C with a ramp of 0.5˚C/min. The fluorescence data following CCD detection was plotted as a function of temperature and fitted to a standard, four parameter sigmoidal equation

$y = LL + \left(\frac{UL-LL}{1+e^{(T_m-T)/a}}\right)$, where y is the observed fluorescence signal, LL and UL are the minimum and maximum intensities in the transition region respectively, 'a' is the slope of the transition, $T_m$ is the melting temperature, and T is the experimental temperature [15].

nanoDSF (Prometheus NT.48) was also used to carry out thermal unfolding experiments of the CcdB mutants. The assays were carried out with 4 μM of each protein and the apparent thermal stability ($T_m$) was determined by monitoring the changes in the fluorescence ratio (F350/F330) as a function of temperature as described earlier [16].

Briefly, the first derivative of the ratio ($F' = \frac{d\frac{F350}{F330}}{dT}$) is normalised using Eq 2:

$$\text{Normalised Fluorescence } (F_N') = \frac{y - y_{min}}{y_{max} - y_{min}} \qquad (2)$$

where y is the first derivative of the observed fluorescence (F350/F330 ratio), $y_{min}$ is value of the first derivative minimum and $y_{max}$ is the value of the first derivative maximum.

A subset of the CcdB mutants were also refolded in 1.5 M GdnCl and subjected to thermal denaturation with native protein in 1.5 M GdnCl as a control. For thermal denaturation, the samples were filled into capillaries and then placed inside the instrument. The capillaries were then heated from 20 to 90˚C with a ramp of 1˚C /min. For the highly stable mutants Y8D, V46L and S60E as well as WT, refolding was carried out in 1, 2, 3, 4 and 5 M GdnCl and refolded protein were subjected to thermal denaturation. Native protein in 1, 2, 3, 4 and 5 M GdnCl was also taken as control.

**TEM-1 β-lactamase, p53-DBD, and mRBD (331–532).** nanoDSF (Prometheus NT.48) was also used to probe the thermal unfolding of the TEM-1 β-lactamase, p53-DBD and mRBD mutants. The assays were carried out with 4 μM of each protein and the apparent thermal stability ($T_m$) was determined by monitoring the changes in the fluorescence ratio (F350/F330) as a function of temperature.

For the TEM-1 β-lactamase, the ratio F350/F330 is normalised using Eq 3:

$$\text{Normalised Fluorescence } (F_N) = \frac{y - y_{min}}{y_{max} - y_{min}} \qquad (3)$$

where y is the observed fluorescence (F350/F330 ratio), $y_{min}$ is value of the ratio minimum and $y_{max}$ is the value of the ratio maximum.

For the p53-DBD, the first derivative of the 330 wavelength ($F330' = \frac{dF330}{dT}$) is normalised using Eq 4:

$$\text{Normalised Fluorescence } (F330_N') = \frac{y - y_{min}}{y_{max} - y_{min}} \qquad (4)$$

where y is the first derivative of the observed fluorescence (F330), $y_{min}$ is value of the first derivative minimum and $y_{max}$ is the value of the first derivative maximum at 330.

For the mRBD mutants, normalization was carried out similar to CcdB mutants using Eq 2.

Samples were heated from 20 to 90˚C with a ramp of 1˚C /min. TEM-1 β-lactamase and p53-DBD were refolded in 0.5 M GdnCl and 0.5 M urea respectively and subjected to thermal

denaturation along with the native proteins in 0.5 M GdnCl and 0.5 M urea as their respective controls. For the highly stable mRBD mutants D389E, L390M and P527I as well as WT, refolding was carried out in 0.5, 1, 2, and 3 M GdnCl and refolded proteins were subjected to thermal denaturation with native protein in 0.5, 1, 2, and 3 M GdnCl as controls.

## Isothermal denaturation of purified proteins

**CcdB.** Briefly, all the CcdB mutant proteins were overnight dialysed four times in total of 2 litres of 200 mM HEPES, pH 8.4 using Tube-O-Dialyser (4kDa MWCO, GBiosciences). The chemical stability ($C_m$) was then determined using 5 μM of proteins by monitoring the changes in the fluorescence ratio (F350/F330), after overnight incubation at 25˚C containing various concentrations of the denaturant (GdnCl). The GdnCl concentrations were estimated from refractive index measurements using a refractometer. The data was analyzed using Sigmaplot for Windows scientific graphing software, and the plots were fitted to a two-state unfolding model ($N_2 \leftrightarrow 2U$). The fraction unfolded for all CcdB mutants was calculated as described [14,16] and is summarized below.

The spectroscopic signal ($Y$) of a protein solution ($F_{350}/F_{330}$ ratio by nano-DSF) is related to the fraction of unfolded protein ($f_u$) by:

$$f_u = \frac{Y - Y_f}{Y_u - Y_f} \tag{5}$$

where $Y_f$ and $Y_u$ are the values of Y for the folded and unfolded protein respectively. These change linearly with the denaturant concentration ([D]), as follows:

$$Y_f = y_f + m_f[D] \tag{6}$$

$Y_u = y_u + m_u[D]$...(7), where $y_f$ and $y_u$ are the folded and unfolded parameters at zero denaturant concentration respectively. $m_f$ and $m_u$ are the denaturation dependence of Y for the folded and unfolded state respectively. $C_m$ is the denaturant concentration at which $f_u = 0.5$. The data are analyzed to yield to obtain the free energy of unfolding at zero denaturant, $\Delta G° = \Delta G_D° - m_{equi}[D]$, where $\Delta G_D°$ is the measured free energy of denaturation at denaturant concentration [D] and $m_{equi}$ encapsulates the denaturant dependence of $\Delta G_D°$ [66]. For homodimeric CcdB, $\Delta G_D°$ is related to $f_u$ by the following equation:

$f_u = \frac{1}{2}\left[-z \pm \sqrt{z^2 + 4z}\right]$, where $z = (e^{-\Delta G_D°/RT})/2P_t$, where $P_t$ is the total protein concentration in terms of monomer units, R is the universal gas constant and T is the absolute temperature [14].

**TEM-1 β-lactamase and p53-DBD.** Equilibrium unfolding experiments of the monomeric TEM-1 β-lactamase and p53-DBD mutants were also carried out by nanoDSF (Prometheus NT.48). The stability to chemical denaturation ($C_m$) was determined by monitoring the changes in the fluorescence ratio (F350/F330), after overnight incubation at 25˚C in the final storage buffer for TEM-1 mutants containing various concentrations of the denaturant (GdnCl) and overnight incubation at 15˚C in the final storage buffer for p53-DBD mutants containing various concentrations of the denaturant (Urea). The data was analyzed using Sigmaplot for Windows scientific graphing software, and the data were fit to a two-state unfolding model (N↔U). Isothermal denaturation was carried out at a fixed protein concentration of 5 μM for all the proteins. The fraction unfolded for all the TEM-1 β-lactamase and p53-DBD mutants were calculated in a similar way as described above.

For all the various classes of proteins used in the study (CcdB, p53-DBD and TEM-1-β-lactamase), the isothermal equilibrium denaturation experiments (for a particular protein) are

performed in the same nanoDSF instrument (Prometheus NT.48) at the same LED settings (100%) and at identical temperatures and buffer conditions for each set of WT and mutant proteins.

## Refolding and unfolding kinetics of purified proteins

**CcdB.** Briefly, the refolding rates were measured using different concentrations (1 μM -5 μM) of the dialysed proteins in 200 mM HEPES, pH 8.4 denatured in 0.2–4 M GdnCl and subsequently diluted to final denaturant concentrations varying from 0.1 M to 1.5 M of GdnCl, and the changes in the fluorescence ratio (F350/F330) were monitored as a function of time. To measure the unfolding kinetics, protein in native buffer (200 mM HEPES, pH 8.4) was diluted into the same buffer containing 8 M GdnCl to a final concentration of GdnCl varying from 2 to 4 M and the changes in the fluorescence ratio (F350/F330) were monitored as a function of time. Refolding kinetic traces of fluorescence intensity from 0.1 M to 1.5 M GdnCl as a function of time for different CcdB mutants were normalized from 0 to 1 between native and denatured baselines, as described previously [8,16,17]. Unfolding kinetic traces of fluorescence intensity from 2 to 4 M GdnCl as a function of time for different CcdB mutants in 200 mM HEPES, pH 8.4 were normalized from 0 to 1 between native and denatured baselines, as described previously [16,17]. For the stable CcdB mutants (Y8D, V46L and S60E as well as WT CcdB) refolding and unfolding were carried out in 2 M and 4.5 M GdnCl respectively in 200 mM HEPES, pH 8.4 at 25˚C and normalisation was done as described above. The data was analyzed using Sigmaplot for Windows scientific graphing software and plots were fitted to a 5 parameter equation for exponential decay for refolding ($y = a0 + (a1 * e^{-kf_1x}) + (a2 * e^{-kf_2x})$), and a 3 parameter exponential rise for unfolding ($y = A0 + (A1 * (1 - e^{-ku_1x}))$) as described previously [17], where x is the time of refolding/unfolding. Both refolding and unfolding studies of all the mutants were also carried out at three different denaturant concentration and the observed rates were plotted as a function of denaturant concentration to determine the refolding and unfolding m values ($M^{-1}s^{-1}$). This was further used to extrapolate the unfolding and refolding rates of all the mutants to 0 M GdnCl concentration for relative comparison as described by the following equation $k_{obs} = k_o + m_{kinetic(\frac{F}{U})}[D]$, where $k_{obs}$ is the observed refolding/unfolding rate constant in $s^{-1}$ at a particular denaturant concentration, $k_o$ is the extrapolated refolding/unfolding rate constant at 0 M denaturant in $s^{-1}$, $m_{kinetic(F)}$, $m_{kinetic(U)}$ is the slope of a plot of the refolding or unfolding rate constant respectively as a function of denaturant concentration in units of $M^{-1}s^{-1}$ and [D] is the denaturant concentration in units of M.

**TEM-1 β-lactamase, p53-DBD and mRBD (331–532).** Refolding and unfolding kinetics of the TEM-1 β-lactamase and mRBD mutants at 25˚C and of the p53-DBD mutants at 15˚C were also monitored by nanoDSF using PR.Time Control software (Prometheus NT.48). Briefly, the refolding rates for the TEM-1 β-lactamase and the mRBD mutants were measured using 5 μM of the proteins denatured in storage buffer containing 3 M GdnCl and subsequently diluted to a final concentration of 0.5 M of GdnCl, and the changes in the fluorescence ratio (F350/F330) were monitored as a function of time. For the p53-DBD mutants, the refolding rates were measured using 4 μM of the proteins denatured in the final storage buffer containing 4 M urea and subsequently diluted to a final concentration of 2 M urea.

To measure the unfolding kinetics, protein in native storage buffer was diluted into buffer containing 2.5 M GdnCl for TEM-1 β-lactamase and 3 M GdnCl for mRBD mutants and 4.4 M Urea for p53-DBD and the changes in the fluorescence ratio (F350/F330) were monitored as a function of time. Refolding and unfolding kinetic traces were normalized from 0 to 1 between native and denatured baselines. The data for the TEM-1 β-lactamase mutants was analyzed using Sigmaplot for Windows scientific graphing software and plots were fitted to a 5 parameter

equation for exponential decay for refolding ($y = a0 + (a1 * e^{-kf_1x}) + (a2 * e^{-kf_2x})$), yielding slow and fast phase rate constants and a 3 parameter exponential rise for unfolding ($y = A0 + (A1 * (1 - e^{-ku_1x}))$) as described above, where x is the time of refolding/unfolding. The data for the mRBD mutants was analyzed using Sigmaplot for Windows scientific graphing software and plots were fitted to a 3 parameter equation for exponential decay for refolding ($y = a0 + (a1 * e^{-kf_1x})$), and a 5 parameter exponential rise for unfolding ($y = A0 + (A1 * (1 - e^{-ku_1x})) + (A2 * (1 - e^{-ku_2x}))$), yielding slow and fast phase rate constants as described above, where x is the time of refolding/unfolding. The data for the p53-DBD mutants was analyzed using Sigmaplot for Windows scientific graphing software and plots were fitted to a 5 parameter equation for exponential decay for refolding ($y = a0 + (a1 * e^{-kf_1x}) + (a2 * e^{-kf_2x})$), yielding slow and fast phase rate constants and a 5 parameter equation for exponential rise for unfolding ($y = A0 + (A1 * (1 - e^{-ku_1x})) + (A2 * (1 - e^{-ku_2x}))$), yielding slow and fast phase rate constants as described above, where x is the time of refolding or unfolding.

For all the various classes of proteins used in the study (CcdB, p53-DBD, TEM-1-β-lactamase and mRBD), the kinetics experiments (for a particular protein) are performed in the same nanoDSF instrument (Prometheus NT.48) at the same LED settings (100%) and at identical temperatures and buffer conditions for each set of WT and mutant proteins.

## Binding of native and refolded CcdB proteins to GyrA14 by MicroScale Thermophoresis (MST)

6.6 μM GyrA14 (in 1X PBS) was labeled using the RED-NHS Monolith Protein Labeling Kit (NanoTemper Technologies) according to the manufacturer's instructions. After labeling, the protein was eluted by gravity flow using a PD MiniTrap G-25 (GE Healthcare) Sephadex column into 200 mM HEPES, pH 8.4 which was also used as assay buffer for MST experiments. The binding of 70 nM of labeled GyrA14 to refolded CcdB mutants in 1.5 M GdnCl (also in 0.1 M GdnCl for V18W, V18W-S12G, V20F and V20F-S12G) as well as native CcdB proteins, not subjected to refolding, in presence of 1.5 M GdnCl (also in 0.1 M GdnCl for V18W, V18W-S12G, V20F and V20F-S12G) was measured. Samples were loaded into Monolith NT.115 MST Standard Capillaries (NanoTemper Technologies) and binding measured using a Monolith NT.115 instrument with MO.Control software at room temperature (LED/excitation power setting 100%, MST power setting 80–100%). Data was analyzed using MO.Affinity Analysis software (version 2.2.5, NanoTemper Technologies) at different standard MST-off times.

## Affinity and thermal tolerance of the CcdB proteins measured by Surface Plasmon Resonance (SPR)

All SPR experiments were performed with a Biacore 2000 (Biacore, Uppsala, Sweden) optical biosensor at 25°C. 35 μg/mL of GyrA14 was used for immobilization at 30 μl/min flow rate for 180 s. 1000 resonance units of GyrA14 were attached by standard amine coupling to the surface of a research-grade CM5 chip. A sensor surface (without GyrA14) that had been activated and deactivated served as a negative control for each binding interaction. The E11R CcdB mutant proteins (E11R, L36A-E11R and L83S-E11R) were overnight dialysed three times against a total 6 litres of 1X PBS, pH 7.4 using Tube-O-Dialyser (4 kDa MWCO, GBiosciences). The remaining proteins were in 200 mM HEPES-0.1 M glycine, pH 8.4. Different concentrations of the dialysed CcdB mutants were run across each sensor surface in a running buffer of PBS (pH 7.4) containing 0.005% Tween surfactant. Protein concentrations ranged from 3 nM to 5 μM. Both association and dissociation were measured at a flow rate of 30 μl/min. In all cases, the sensor surface was regenerated between binding reactions by one to two

washes with 4 M $MgCl_2$ for 10–30 s at 30 μL/min. Each binding curve was corrected for non-specific binding by subtraction of the signal obtained from the negative control flow cell. The kinetic parameters were obtained by fitting the data to a simple 1:1 Langmuir interaction model by using BIA EVALUATION 3.1 software as described previously [8,67]. Thermal tolerance of the CcdB mutants E11R, S12G, L36A, L36A-E11R and L36A-S12G and WT CcdB using 500 nM of protein was assessed by their ability to bind GyrA14 after heat stress. The protein sample was incubated at 40˚C and 80˚C respectively for 1 hour in a PCR cycler (BioRad) with a heated lid to prevent evaporation. The samples were cooled to 25˚C and binding affinity to GyrA14 was determined by SPR experiments as described above The fraction of active protein following thermal stress was quantitated by measuring the RUs at the end of the association time period relative to those for the same protein incubated throughout at 25˚C. This was designated as Residual activity.

## MIC and $IC_{90}$ determination of TEM-1 β-lactamase mutants

Cultures were grown overnight at 37˚C in LB broth with 50 μg/mL kanamycin. The overnight cultures were diluted $1:10^4$ and 30 μL was inoculated into 500 μL of fresh LB broth supplemented with 50 μg/mL kanamycin, 100 μM IPTG inducer, and various concentrations of ampicillin and cefotaxime. The concentrations of ampicillin used for MIC determination were 0, 25, 50, 100, 200, 300, 400, 500, 600, 800, 1000, 1500, 2000, 2500, 3000 and 4000 μg/mL. The concentrations of cefotaxime tested were 0, 1, 2, 3, 4, 5, 6, 8, 10, 15, 20 and 30 μg/mL. The cultures were then incubated at 37˚C with shaking for 24 hours following which $OD_{600}$ measurements were carried out on Varioskan Flash (ThermoScientific) using Nunclon delta surface plates (ThermoScientific). The MIC was determined by recording the lowest concentration of ampicillin or cefotaxime on which no growth was observed. In practice, this was an all or none phenomenon [24].

$IC_{90}$ was derived directly from the plate data measurements. Briefly, *E. coli* BL-21 (λDE3, plysE) containing the pET24a plasmid that encodes TEM-1 β-lactamase mutants was grown overnight in LB broth with 50 μg/mL kanamycin. Overnight cultures were diluted 1:100 into LB broth with 50 μg/mL kanamycin and incubated for 4 hrs at 37˚C to mid-log phase ($OD_{600}$ ~0.6). Ten-fold serial dilutions of each culture were made, and 100 μL of each dilution was spread onto LB agar plates containing 100 μM IPTG inducer and 0–4000 μg/mL ampicillin and 0–50 μg/mL of cefotaxime, in a series of two-fold increments. After incubation for 24 hrs at 37˚C, colony forming units (cfu) on each plate were counted to calculate the cfu/mL of culture, and $IC_{90}$ was defined as the concentration of ampicillin or cefotaxime that reduces cfu/mL of culture by ≥90% [22].

**Nitrocefin assay of TEM-1 β-lactamase mutants.** TEM-1 β-lactamase activity was assayed as described previously [6]. Briefly the rate of nitrocefin (50 μM) hydrolysis was observed at 486 nm at 25˚C for 60 minutes in 10 mM HEPES, 300 mM NaCl, 10% glycerol (pH 7.0) using 10 nM native protein, native protein in 0.5 M GdnCl and refolded protein in 0.5 M GdnCl. Activity measurements were carried out on Varioskan Flash (ThermoScientific) using Nunclon delta surface plates (ThermoScientific).

**SPR-binding of native, native in presence of GdnCl and refolded mRBD proteins with immobilized ACE2-hFc.** Binding studies of various mRBD mutants with ACE2-hFc neutralizing antibody were carried out using the ProteOn XPR36 Protein Interaction Array V.3.1 from Bio-Rad as described previously [34]. Briefly, following activation of the GLM sensor chip with EDC and sulfo-NHS (Sigma), Protein G (Sigma) was coupled at 10 μg/mL in the presence of 10 mM sodium acetate buffer pH 4.5 at 30 μL/min for 300 seconds until ~3500–4000 RU was immobilized. After quenching the excess sulfo-NHS esters using 1 M

ethanolamine, ~1000 RU of ACE2-hFc was immobilized at a flow rate of 5 μg/mL for 100 seconds on various channels except one blank channel that served as the reference channel. Native mRBD proteins, proteins in 0.5 M GdnCl and refolded protein in 0.5 M GdnCl were passed at a flow rate of 30 μL/min for 200 seconds over the chip surface, followed by a dissociation step of 400 seconds. A lane without any immobilization was used to monitor non-specific binding. After each kinetic assay, the chip was regenerated in 0.1 M Glycine-HCl (pH 2.7). 50 nM of the mRBD proteins in 1X PBS were used in all cases for the binding studies. The response units for each of the native protein, native protein in 0.5 M GdnCl and refolded proteins in 0.5 M GdnCl was used for relative comparisons. For the studies carried out in GdnCl, the running buffer did not have any GdnCl and the jumps obtained in all the channels were removed after reference subtraction.

## Crystallization of CcdB mutants

5 mg/ml of the purified mutants of CcdB, S12G, V46L and S60E, in 1X PBS pH 7.4 and 1 mM EDTA were screened for crystallization with Hampton Research screens by the sitting drop method using the Mosquito crystallization robot facility at NCBS/Instem. Plates were incubated at 18˚C. Crystals appeared in a few conditions after approximately 20 days. Two conditions had mountable crystals: 0.2 M calcium chloride dihydrate, 20% w/v PEG 3350, pH 5.1 (PEG/Ion 1, condition 7) and 0.2 M ammonium chloride, 20% w/v PEG 3350, pH 6.3 (PEG/Ion 1, condition 9). The initial crystals diffracted to ~2.5 Å. Conditions were further optimized to obtain single crystals for better diffraction quality. The best crystals were obtained in the condition 0.15 M calcium chloride dihydrate, 20% w/v PEG 3350 for S12G; 0.20 M calcium chloride dihydrate, 25% w/v PEG 3350, pH 5.1 with 20% glycerol (cryo) for V46L; 0.20 M calcium chloride dihydrate, 10% w/v PEG 3350; pH 5.1 with 20% glycerol (cryo) for S60E, with a protein:precipitant ratio of 1:2 at 18˚C using the hanging drop method.

## Data collection and processing of CcdB mutants

Diffraction data was collected at 100K using Rigaku FR-X with R-AXIS IV++ detector facility at NCBS/Instem for S12G. For V46L, diffraction data was collected at 100K using XRD2 beamline with Dectris Pilatus-6M detector facility at Elettra synchrotron, Trieste, Italy. For S60E, diffraction data was collected at 100K using Rigaku MicroMax-007HF with mar345dtb detector facility at home-source MBU, IISc. The crystals diffracted to 1.63Å, 1.35 Å and 1.93 Å for S12G, V46L and S60E respectively. Data was processed using iMOSFLM [68,69] with an overall completeness of 90.6% for S12G, 100% for V46L and 97.5% for S60E. For S12G, the crystal belonged to the C2 space group, with the unit cell parameters, while the pointless predicted I2. Thus, the data was re-scaled in I2 with unit cell parameters a = 35.57; b = 36.53; c = 67.53 and β = 93.69˚. The V46L and S60E crystals belonged to the C2 space group with unit cell parameters a = 75.10; b = 36.76; c = 35.91 and β = 115.19˚ for V46L and a = 74.81; b = 36.64; c = 35.67 and β = 114.97˚ for S60E. The structure was solved by molecular replacement using PHASER [70] with 3VUB [71] as the starting model. After iterative cycles of refinement using Refmac [72,73] and manual model building using Coot [74,75], the final model consisted of 816 protein atoms, 95 water molecules and two Cl⁻ ions with an $R_{factor}$ and $R_{free}$ of 22.8 and 26.9 respectively for S12G. For V46L, the final model consisted of 814 protein atoms, 167 water molecules and two Cl⁻ ions with an $R_{factor}$ and $R_{free}$ of 16.2 and 18.4 respectively. For S60E, the final model consisted of 820 protein atoms, 98 water molecules and two Cl⁻ ions with an $R_{factor}$ and $R_{free}$ of 17.0 and 21.4 respectively. Composite omit maps were calculated around the 12[th], 46[th] and 60[th] residue for S12G, V46L and S60E respectively. The final model was validated using the validation server (https://validate.wwpdb.org). Data processing and refinement

statistics are given in S10 Table. The server "https://swift.cmbi.umcn.nl/servers/html/listavb.html" was used for calculating the average B-factors.

## Analysis of expression and GyrA14 binding of CcdB mutants on the yeast cell surface

Plasmids expressing Aga2p fusions of L42E, inactive mutants (V5F, V18W, V20F, L36A, L83S), the double mutants of L42E made in the background of the inactive mutants, and WT CcdB were transformed into *Saccharomyces cerevisiae* EBY100 cells as described [35]. Briefly, the amount of CcdB protein expressed on the yeast cell surface was estimated by chicken anti-HA antibodies from Bethyl labs (1:600 dilution) and the GyrA14 binding activity on the yeast cell surface was estimated by incubating the induced CcdB mutants with 100 nM FLAG tagged GyrA14, followed by washing with FACS buffer and subsequent incubation with mouse anti-FLAG antibodies, at a dilution ratio of 1:300 as described previously [35]. This was followed by washing the cells twice with FACS buffer, and incubating with goat anti-chicken antibodies conjugated to Alexa Fluor 488 (1:300 dilution) for expression, and rabbit anti-mouse antibodies conjugated to Alexa Fluor 633 (1:1600 dilution) for binding, for 20 minutes at 4˚C. The flow cytometry was performed using a BD Aria III instrument.

**Statistical analysis.** All the experiments are carried out in biological replicates (n = 2), and the listed errors are the standard errors derived from the values obtained for individual replicates. For the nanoDSF and MST measurements, each experiment has been carried out twice, each time with two sets of capillaries (n = 4) and the listed errors are the standard errors derived from the values obtained for individual replicates. The SPR experiments in S4 Fig have been performed once at each concentration with four different concentrations of the protein and the listed error is the standard error derived from the values at multiple concentrations. The P values for comparing the kinetic parameters, were analysed with a two-tailed Mann Whitney test using the GraphPad Prism software 8.0.0 (* indicates P < 0.05, ** indicates P < 0.005, *** indicates P < 0.0005).

## Supporting information

**S1 Fig. Enhancement of solubility and stability by the suppressors (related to Fig 1). (A-B) Solubility and thermal stabilities of CcdB mutants, in presence and absence of the suppressors.** (A) *In vivo* solubility estimates for CcdB mutants. UI, T, S and P are uninduced, total cell lysate, supernatant and pellet respectively. Ś is the purified CcdB WT protein used as standard and M is the molecular weight marker lane. The relative estimates of protein present in the soluble fraction and inclusion bodies for all mutants are shown in S1 Table. The red arrow indicates the band for the induced protein. (B) Thermal unfolding profiles of purified WT CcdB and CcdB mutants in the absence and presence of 8 μM CcdA peptide (45–72) measured by a thermal shift assay (TSA). L83S-E11R and V20F are omitted in the top and bottom panels as they do not show clear thermal transitions in the absence and presence of CcdA respectively. (TIFF)

**S2 Fig. Denaturant-dependent refolding and unfolding kinetics of CcdB mutant proteins (related to Fig 2).** (A-E) Representative refolding rate constants of fast phase (left side) and slow phase (right panel) and (F-H) unfolding rate constants at 5 μM protein concentration of the WT and mutants are shown. The experimental rate constants (for refolding and unfolding) obtained at increasing final GdnCl concentrations are shown in dots, while fits are shown in solid lines. For refolding kinetics, the extrapolated rate constants and the magnitude of refolding m-values of the transition states at zero denaturant concentration of fast phase (B and D

respectively) and slow phase (C and E respectively) are shown. Suppressor mutations significantly accelerate the refolding rate constants. For unfolding kinetics, the extrapolated rate constants (G) and the magnitude of unfolding m-values (H) at zero denaturant concentration are shown. Suppressor mutations decrease the unfolding rate constants. The error bars wherever shown represent the standard deviation from two independent experiments, each performed in duplicates (see also S4 Table).
(TIFF)

**S3 Fig. Interaction of native, native in the presence of GdnCl and refolded CcdB WT and mutant proteins with labeled GyrA14 analyzed by MicroScale Thermophoresis (related to Fig 2).** GyrA14 was labeled and used at a concentration of 70 nM and titrated with different concentrations of native (A), native in the presence of GdnCl (B) and refolded (C) WT and CcdB mutants. All studies were carried out in 200 mM HEPES, pH 8.4, at 27°C. The normalised fluorescence FNorm [$^0/_{00}$] is plotted as a function of [CcdB]. For each capillary (each measuring point), an MST trace is recorded. All traces are then normalised to start at 1000. For each trace, the FNorm value for the dose-response curve is calculated from Fhot (MST laser on)/Fcold (MST laser off). The dissociation constants ($K_D$) listed in S1 Table, were determined employing standard data analysis with MO.Affinity Analysis Software (1). (D) Equilibrium GdnCl denaturation profile of 5 μM of GyrA14 carried out in 1X PBS, pH 7.5 at 25°C using nanoDSF. The experimental data is shown in black dots, while the fit is shown in a red line. The $C_m$ of GyrA14 is 4.48 M thus proving it to be stable and folded in the MST studies carried out in the presence of 1.5 M GdnCl.
(TIFF)

**S4 Fig. Binding of GyrA14 to CcdB proteins (related to Fig 2).** Overlays show the binding kinetics with analyte concentration increasing from the bottom to the top curve in all cases of (A) WT CcdB (3, 6, 12.5, 25 nM); (B) S12G (3, 6, 25, 50 nM); (C) E11R (50, 100, 200, 500 nM); (D) V18W (200, 500, 1000, 2000 nM); (E) V18W-S12G (200, 500, 1000, 2000 nM); (F) V20F (200, 1000, 2000, 5000 nM); (G) V20F-S12G (100, 200, 500, 1000 nM); (H) L36A (12.5, 25, 100, 200 nM); (I) L36A-S12G (3, 6, 12.5, 25 nM); (J) L36A-E11R (10, 21.4, 32, 64.2 nM); (K) L83S (200, 500, 1000, 2000 nM); (L) L83S-S12G (200, 500, 1000, 2000 nM); (M) L83S-E11R (50, 100, 200, 500 nM). The ligand GyrA14 was immobilized on the CM5 chip by standard amine coupling. Binding was measured by passing varying concentrations of the analyte (CcdB proteins) over the ligand (GyrA14) immobilised chip and the data was fitted to the 1:1 Langmuir Interaction model to obtain the kinetic parameters (S1 Table).
(TIFF)

**S5 Fig. Refolding and unfolding kinetics, and hydrolysis activities of TEM-1 β-lactamase mutant proteins (related to Fig 3) as well as refolding and unfolding kinetics of p53-DBD mutant proteins (related to Fig 4).** (A) TEM-1 proteins exhibit biphasic refolding kinetics with a fast and slow phase whereas (B) unfolding follows single exponential kinetics. Representative kinetic traces at 5 μM protein concentrations of WT and TEM-1 mutants are shown in presence of 0.5 M GdnCl for refolding and 2.5 M GdnCl for unfolding. The experimental kinetic traces obtained at the indicated GdnCl concentrations are shown in black, while the fits are shown in red. The measured kinetic parameters are listed in S6 Table. The lactamase activities of the WT and mutants in the following conditions–(C) native, (D) native proteins in 0.5 M GdnCl and (E) refolded proteins in 0.5 M GdnCl, were assayed by observing the rate of nitrocefin hydrolysis (50 μM) at 486 nm at 25°C at a protein concentration of 10 nM. The M182T suppressor rescues the activity of M69I and L76N mutants. (F) p53-DBD proteins exhibit biphasic refolding kinetics with significant burst, fast and slow phases whereas (G)

unfolding of p53-DBD follows biphasic exponential kinetics with burst, fast and slow phases. Representative kinetic traces at 5 µM protein concentration of WT and p53-DBD mutants are shown in the presence of 2 M Urea for refolding and 4.4 M Urea for unfolding. The experimental kinetic traces obtained are shown in black, while the fits are shown in red. The measured kinetic parameters are listed in S7 Table.
(TIFF)

**S6 Fig. Refolding and unfolding kinetics, and thermal denaturation studies of stabilised CcdB and mRBD proteins with suppressor mutations; binding studies of the mRBD proteins to ACE2-hFc neutralizing antibody (related to Fig 5).** (A) Biphasic refolding kinetics with a fast and slow phase and (B) single exponential unfolding kinetics of CcdB mutant proteins. Representative kinetic traces at 5 µM protein concentration of WT CcdB; Y8D; V46L; S60E are shown. (C) mRBD proteins exhibit single phase refolding kinetics with a significant burst phase whereas (D) unfolding of mRBD proteins follows biphasic exponential kinetics with burst, fast and slow phases. Representative kinetic traces at 5 µM protein concentration of WT mRBD; D389E; L390M; P527I are shown. The experimental kinetic traces obtained at different GdnCl concentrations are shown in black, while the fits are shown in red. The measured kinetic parameters are listed in S9 Table for CcdB and mRBD mutants. (E-H) Thermal denaturation traces of 5 µM of native CcdB proteins in (E) 1 M, (F) 2 M, (G) 3 M and (H) 4 M GdnCl (represented by dashed lines) and refolded CcdB proteins in the same concentrations of GdnCl (represented by solid lines). WT CcdB failed to show any transition at 3 M, whereas Y8D and V46L failed to show a thermal unfolding transition at 4 M. S60E showed a thermal transition at 4 M. (I-K) Thermal denaturation traces of 5 µM of native mRBD proteins in (I) 0.5 M, (J) 1 M and (K) 2 M (represented by dashed lines) and refolded mRBD proteins in same concentrations of GdnCl (represented by solid lines). mRBD WT failed to show any transition at 2 M, whereas the stabilised mutants show thermal unfolding upto 2M GdnCl. (L-O) Overlays show the ACE2-hFc binding of the WT and mRBD mutants in the following conditions–native, native in 0.5 M GdnCl and refolded in 0.5 M GdnCl for (L) WT mRBD, (M) D389E, (N) L390M and (O) P527I. The ligand ACE2-hFc was immobilized on Protein-G immobilised GLM sensor chip. Binding was measured by passing 50 nM of the analyte (mRBD proteins) over the ligand (ACE2-hFc) immobilised chip.
(TIFF)

**S7 Fig. Conformational change of the 8–15 loop ([8]YKRESRYR[15]) in the CcdA bound structure.** The crystal structures of CcdB in (A) free (PDB ID:3VUB) and (B) CcdA bound state (PDB ID:3G7Z) are shown (2,3). The CcdB dimer is shown in cyan and the 8–15 loop and 39–52 loop are shown in blue and red respectively. The CcdA dimer is shown in green in the CcdA bound state. The conformation of the residues 8–15 and 39–52 change in the CcdA bound state as most of these residues of these loops are involved in direct interaction with CcdA.
(TIFF)

**S8 Fig. Fold change of different kinetic parameters of the folding process in various protein systems (Related to Fig 8).** (A-C) $\log_2$ fold change of various kinetic parameters (Mean ±SEM) for the suppressor mutations in the background of WT or inactive mutant for (A) CcdB, (B) TEM-1 β-lactamase and (C) mRBD proteins. Mann Whitney non-parametric test was performed for each of these parameters. The mean of the distributions of the values for each of the parameters are significantly higher than $\log_2(2)$ for refolding. P value indicated with *, ** and *** indicates $< 0.05$, $< 0.005$ and $< 0.0005$ respectively. $a0$, $kf_1$, $kf_2$, $A0$, $ku_1$, $ku_2$ are the burst phase amplitude for refolding, rate constant of fast phase of refolding, rate

constant of slow phase of refolding, amplitude of burst phase of unfolding, rate constant of fast phase of unfolding, and the rate constant of slow phase of unfolding respectively.
(TIF)

**S9 Fig. Detailed intramolecular interactions between residues in three crystal structures of CcdB suppressor mutants, namely, (A) S12G, (B) V46L and (C) S60E (Related to Fig 6).** The CcdB chains are coloured in yellow in S12G, magenta in V46L and green in S60E, while the interacting residues are coloured in blue. Additional hydrogen-bonded interactions present in the suppressor, but absent in the WT (blue dotted lines) between polar and charged residues are shown.
(TIFF)

**S1 Table. Fractional solubilities, thermal stabilities (determined by nanoDSF) and GyrA14 binding affinities (determined by SPR and MST) of different CcdB mutants[1] (Related to Figs 1 and 2).** Binding affinities of refolded proteins and proteins in GdnCl were also measured by MST. SPR experiments were carried out at 25˚C, pH 7.4 and MST experiments were carried out at 25˚C, pH 8.4. [1]Reported standard errors are derived from two independent experiments, each performed in duplicates. [2]Mutants could not be purified. [3,4] Refolded in 0.1 M GdnCl. All the other proteins were refolded in 1.5 M GdnCl. NB: No Binding.
(DOCX)

**S2 Table. Thermodynamic stability parameters ($C_m$, $\Delta G^0$, $m_{equi}$), following chemical denaturation and melting temperatures of refolded and native proteins in presence of 1.5 M GdnCl ($T_{mRefold}$, $T_{mGdnCl}$) determined by nanoDSF, of different CcdB mutants[1] (Related to Fig 2).** Residual activity after high temperature incubation of CcdB mutants was determined by SPR at room temperature. [1]Reported standard errors are derived from two independent experiments, each performed in duplicates. [2,3] Could not be refolded in 1.5 M GdnCl. All the other proteins were refolded in 1.5 M GdnCl.-Not determined.
(DOCX)

**S3 Table. Kinetic parameters for refolding and unfolding of CcdB mutants measured under different buffer conditions at 25˚C, pH 8.4[1] (Related to Fig 2).** [1]Reported standard errors are derived from two independent experiments, each performed in duplicates.
(DOCX)

**S4 Table. Relative kinetic parameters for *in vitro* refolding and unfolding of CcdB mutants extrapolated to zero denaturant[1] (Related to Fig 2).** [1]Reported standard errors are derived from two independent experiments, each performed in duplicates. [2]Reported rate constants are extrapolated to 0 M GdnCl for relative comparison among different mutants. [3] Fast phase of refolding couldn't be captured.
(DOCX)

**S5 Table. Top: MIC and $IC_{90}$ for ampicillin and cefotaxime from $OD_{600}$ and plate data measurements respectively, thermodynamic stability parameters ($C_m$, $\Delta G^0$, $m_{equi}$), apparent thermal stability ($T_m$), thermal stabilities of refolded proteins and native proteins in presence of 0.5 M GdnCl ($T_{mRefold}$, $T_{mGdnCl}$), of different TEM-1 β-lactamase mutants[1]. Bottom: Kinetic parameters for refolding and unfolding of TEM-1 β-lactamase mutants measured in 0.5 M and 2.5 M GdnCl respectively carried out in 10 mM HEPES, 300 mM NaCl, 10% glycerol, pH 7.0, at 25˚C[1] (Related to Fig 3).** [1]Reported standard errors are derived from two independent experiments, each performed in duplicates. NT- No Transition: Not refolded in 0.5 M GdnCl. All the other proteins were refolded in 0.5 M GdnCl.
(DOCX)

**S6 Table. Top: Fractional solubilities, thermodynamic stability parameters ($C_m$, $\Delta G^0$, $m_{equi}$), apparent thermal stabilities ($T_m$), thermal stabilities of refolded and native proteins in the presence of 0.5 M Urea ($T_{mRefold}$, $T_{m,urea}$) determined by nanoDSF, of different p53 mutants[1]. Bottom: Kinetic parameters for refolding and unfolding of p53 mutants measured in 2 M and 4.4 M Urea respectively carried out in 50 mM $NaH_2PO_4$, 500 mM NaCl, 500 mM imidazole, 10 mM DTT, pH 7.2 at 15˚C[1] (Related to Fig 4).** [1]Reported standard errors are derived from two independent experiments, each performed in duplicates. ND–Not determined as the corresponding proteins could not be purified.
(DOCX)

**S7 Table. Thermodynamic parameters ($C_m$, $\Delta G^0$, $m_{equi}$), determined by nanoDSF, of CcdB mutants[1] (Related to Fig 5).** [1]Reported standard errors are derived from two independent experiments, each performed in duplicates. WT values are taken from S5 Table.
(DOCX)

**S8 Table. Top: Kinetic parameters for refolding and unfolding of stabilised CcdB mutants measured in 2 M and 4.5 M GdnCl respectively, carried out in 200 mM HEPES, pH 8.4 at 25˚C[1]. Bottom: Kinetic parameters for refolding and unfolding of stabilised mRBD mutants in 0.5 M and 3 M GdnCl respectively, carried out in 1X PBS, pH 7.0 at 25˚C[1] (Related to Fig 5).** [1]Reported standard errors are derived from two independent experiments, each performed in duplicates.
(DOCX)

**S9 Table. Top: Thermodynamic parameters ($C_m$, $\Delta G^0$, $m_{equi}$), apparent thermal stability ($T_m$), thermal stabilities of refolded and native proteins in presence of 1.5 M GdnCl ($T_{mRefold}$, $T_{mGdnCl}$). Bottom: Kinetic parameters for refolding and unfolding of CcdB WT, M32T, L42E and S43T in 1.5 M and 3.5 M GdnCl respectively carried out in 200 mM HEPES, pH 8.4 at 25˚C[1] (Related to Fig 7).** [1]Reported standard errors are derived from two independent experiments, each performed in duplicates. WT values of thermodynamic parameters are taken from S5 Table and kinetic parameters are taken from S2 Table.
(DOCX)

**S10 Table. Data collection and refinement statistics for CcdB mutants (Related to Fig 6).** *: Values within brackets are for highest resolution shell.
(DOCX)

**S11 Table. Average difference and fold change of various thermodynamic and kinetic parameters (Mean±SEM) respectively for the suppressor mutations averaged over data for mutants of CcdB, TEM-1 β-lactamase and mRBD (Related to Fig 8).** $\Delta\Delta G^\circ$, $\Delta T_m$, a0, $kf_1$, $kf_2$, A0, $ku_1$, $ku_2$ represent the difference in the Gibbs free energy, difference in melting temperature, fold change of amplitude of burst phase of refolding, rate constant of fast phase of refolding, rate constant of slow phase of refolding, amplitude of burst phase of unfolding, rate constant of fast phase of unfolding, rate constant of slow phase of unfolding respectively.
(DOCX)

**S12 Table. Average fold change of various kinetic parameters (Mean±SEM) respectively for the suppressor mutations in the individual proteins CcdB, TEM-1 β-lactamase and mRBD (Related to S8 Fig).** n.a. not applicable as this phase is absent. a0 –amplitude of burst phase of refolding process. $kf_1$, $kf_2$ –refolding rate constants of fast and slow phases respectively. A0 – amplitude of burst phase of unfolding process. $ku_1$, $ku_2$ – unfolding rate constants

of fast and slow phases respectively.
(DOCX)

**S13 Table. Additional Hydrogen bonds present in structures of suppressor mutants but absent in WT, calculated using HBPLUS (Related to S9 Fig). [1]A:** Chain ID, **SC:** Side Chain, **MC:** Main Chain
(DOCX)

**S14 Table. Difference in Accessible Surface Area (ΔASA) between mutant and WT CcdB structures calculated using NACCESS (Related to S9 Fig).** Only residues with $|\Delta ASA|>10\text{Å}^2$ are shown. [1]$\Delta ASA = $ (Total side chain ASA)$_{Mutant}$ – (Total side chain ASA)$_{WT}$.
(DOCX)

## Acknowledgments

We are thankful to Dr. Sivaramaiah Nallapeta and Dr. Saji Menon, for use of the nano-DSF and MST facilities. Prof. S. Ramaswamy, Senior Professor, InStem is acknowledged for the use of the X-ray crystallography facility for the diffraction data collection. We also acknowledge both the Elettra synchrotron facility and home source, MBU, IISc for diffraction data collection. The NCBS and MBU, IISc X-ray Facilities were supported by DBT Grant BT/PR5081/INF/156/2012 and DST-SERB grant IR/SO/LU-003/2010 respectively. We thank the beamline staff at the Elettra XRD2, particularly Dr. Raghurama Hegde for beamline support. Access to the XRD2 beamline at the Elettra synchrotron, Trieste was made possible through a grant-in-aid from the DST, India, [grant number DSTO-1668]. We thank Dr. Mahavir Singh, IISc for providing the BL21-Rosetta (DE3) strain. We thank Mohammad Suhail Khan for transfection of the mRBD mutants, and Nonavinakere Seetharam Srilatha, Kawkab Kanjo for assistance with SPR and ProteOn Studies respectively. We thank Dr. Arvind Penmatsa and Arunabh Athreya (MBU, IISc), for helping with the data collection at the synchrotron facility. We thank Munmun Bhasin for helping with HBPLUS and NACCESS calculations. We also thank all the members of the RV lab for their valuable suggestions.

## Author Contributions

**Conceptualization:** Gopinath Chattopadhyay, Jayantika Bhowmick, Raghavan Varadarajan.

**Data curation:** Gopinath Chattopadhyay, Jayantika Bhowmick, Kavyashree Manjunath, Shahbaz Ahmed, Parveen Goyal.

**Formal analysis:** Gopinath Chattopadhyay, Jayantika Bhowmick, Kavyashree Manjunath, Parveen Goyal, Raghavan Varadarajan.

**Funding acquisition:** Raghavan Varadarajan.

**Investigation:** Gopinath Chattopadhyay, Jayantika Bhowmick, Raghavan Varadarajan.

**Methodology:** Gopinath Chattopadhyay, Jayantika Bhowmick, Kavyashree Manjunath, Parveen Goyal, Raghavan Varadarajan.

**Project administration:** Raghavan Varadarajan.

**Resources:** Raghavan Varadarajan.

**Software:** Kavyashree Manjunath, Parveen Goyal.

**Supervision:** Raghavan Varadarajan.

**Validation:** Gopinath Chattopadhyay, Jayantika Bhowmick, Kavyashree Manjunath, Parveen Goyal, Raghavan Varadarajan.

**Visualization:** Gopinath Chattopadhyay, Jayantika Bhowmick, Kavyashree Manjunath, Raghavan Varadarajan.

**Writing – original draft:** Gopinath Chattopadhyay, Jayantika Bhowmick, Kavyashree Manjunath, Raghavan Varadarajan.

**Writing – review & editing:** Gopinath Chattopadhyay, Jayantika Bhowmick, Kavyashree Manjunath, Shahbaz Ahmed, Parveen Goyal, Raghavan Varadarajan.

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
