## [Decision Letter · Decision Letter 0]

9 Jun 2022

Dear Dr Varadarajan,

Thank you very much for submitting your Research Article entitled 'Mechanistic
insights into global suppressors of protein folding defects' to PLOS Genetics.

The manuscript was fully evaluated at the editorial level and by independent peer
reviewers. The reviewers appreciated the attention to an important topic but
identified some concerns that we ask you address in a revised manuscript

We therefore ask you to modify the manuscript according to the review
recommendations. Your revisions should address the specific points made by each
reviewer.

[LINK]

Yours sincerely,

Diarmaid Hughes

Associate Editor

PLOS Genetics

Josep Casadesús

Section Editor: Prokaryotic Genetics

PLOS Genetics

Reviewer's Responses to Questions

**Comments to the Authors:**

Reviewer #1: This manuscript describes the suppressor effects of second mutations on
the defect of functions of proteins by first mutation. The authors performed various
experiments including equilibrium and kinetics analyses of protein folding and
derived conclusions that the significant factor is a refolding rate, and the
stability of the native structure of a protein is not important. The topic authors
treated is interesting and significant in terms of the evolution of protein
functions. Thus, I feel that this manuscript can be basically published but the
following points should be reconsidered before publication.

I think that the less significance of stability of a native structure for suppressor
effects is plausible. But, the significance of refolding rate for suppressor effects
is difficult to understand. What process of refolding is significant? The authors
should discuss this point. Of course, the definite factor of this phenomenon may be
difficult to demonstrate at this stage, but at least the authors should speculate
this point.

It looks like that the native state of a suppressor mutant is less stable but its
refolding rate becomes large as shown in the results of TYEM-1 b lactamase. How
these phenomena can be interpreted in terms of mutations? Which mutations make the
native state unstable and which mutations stabilize the folding transition state?
The authors should give some speculations. The similar analyses should be given for
other proteins.

Minor point;

In page 16, line 407, should “Figure 7G” be “Figure 7D”?

In page 16, line 410, should “Figure 7G” be “Figure 7E”?

In page 18, line 467, the conformation of from residue 9 to 15 (9KRESRYR15) should be
presented in a figure.

Reviewer #2: This manuscript describes a remarkably thorough characterization of
global suppressors in CcdB, (a toxin involved in the maintenance of F-plasmid in
Escherichia coli) together with three other well-studied proteins. CcdB (Controller
of Cell Death protein B) is particularly well characterized using a protocol that
selects for a population of inactive mutants and then selects for global suppressors
within this population.

Mutations are identified that stabilize the WT protein but fail to act as global
suppressors, and the results provide persuasive support for an overall conclusion
that "thermodynamic stabilization is neither necessary nor sufficient for
suppression." Rather, as summarized in the Discussion, "the findings collectively
indicate the role of kinetic stability, in particular an increase in the refolding
rate constant as being primarily responsible for the suppressor phenotype."

The authors are seeking an underlying mechanism that explains global suppression, an
issue of fundamental importance. Indeed, the finding that enhancement of refolding
rates in these suppressors is important, but I don't think it is a "mechanism" in
the usual sense of the word. The attendant finding that inactivating mutants largely
affect the core while global suppressors localize at the solvent-accessible surface
(especially in loops) is both intriguing and suggestive.

What is it about distal loops that can affect global suppression? For what it's
worth, an old result from John Gerlt comes to mind. Here, deletion of an entire loop
in staphylococcal nuclease resulted in a protein that is significantly more active
than wild type. [Poole et al (1991) Deletion of the omega-loop in the active site of
staphylococcal nuclease. 1. Effect on catalysis and stability, Biochemistry 30,
3621-3627. Baldisseri et al (1991) Deletion of the omega-loop in the active site of
staphylococcal nuclease. 2. Effects on protein structure and dynamics, Biochemistry
30, 3628-3633.]

A question that arises is the interaction between the CcdB loop that includes
suppressor candidate residues 10-12, and cognate structures, such as CcdA. The
authors do raise this issue, but I would have welcomed slightly more structural
information about the extent to which protein:protein binding might complicate the
interpretation of suppression. Along similar lines, the high resolution structures
of S12G, V46L and S60E seem to invite detailed assessment of differences in
hydrogen-bonding, solvent accessibility, and possible intramolecular interactions
between proximal loops, beyond those shown pictorially in fig. 6.

To show I read the paper carefully:

lines 294, 296: fold, not "folds", e.g., 10-fold, not "10-folds".

line 488: To, not "Till"

line 502: delete final comma before the period.

This is an outstanding paper

Reviewer #3: In this work Varadarajan and coworkers explore biophysical mechanisms by
which global suppressors act on several proteins whose knockout mutations they
suppress.

The analysis focuses on two key biophysical factors – thermal stability and folding
kinetics using several proteins from various sources and therapeutic areas as
example. The authors find that while suppressor mutations often increase stability
to compensate for destabilization by detrimental mutations this is not always the
case and they note a number of exceptions. In terms of kinetics the authors see a
more consistent trend whereby suppressor mutations speed up in vitro folding
kinetics both in fast and slow phase and they attribute the rescue effect primarily
to kinetic rather than thermodynamic stabilization though direct link between
kinetic stabilization and functional effect of suppressors in vivo has not been
established in this work. However in discussion they speculate that somehow kinetic
stabilization might lead to increase in abundance of functional proteins in the
cell, perhaps through the effect on cotranslational folding mediated by non-ATP
dependent chaperones.

The paper is interesting and the analysis is careful and comprehensive and I
recommend publication with enthusiasm after the following concerns are
considered:

1) The authors emphasize that effect on kinetic rather than thermodynamic stability
determine suppressor mutations. Figure 8 is supposed to convey this message but its
presentation is cursory, even most notations are not defined and it doe not impress
on the reader that thermodynamic effects of suppressors are less significant than
kinetic ones. E.g. amplitudes a_0 and A_0 are not defined and presentation of Fig.8
appears only in the Discussion. I am not sure therefore how Fig.8 conveys main
message of the paper.

2) In Fig.1 kinetic traces are provided but no analysis of rates, amplitudes etc are
given

3) The authors suggest that the effect of mutations on vitro kinetics may indicate a
more profound effect on cotranslational folding and resulting yields of functional
proteins. Recent paper PMID: 31911473 provides evidence of direct link between
kinetic traps manifest in biphasic folding kinetics in vitro and effects on
cotranslational folding. In this sense it may be worthwhile to explore the linear
association between PIM and/or loci of rescue mutations and slow (for E. coli)
codons in endogenous Ec genes such as ccdB

4) The authors point out to examples of stabilizing rescue mutations. However, one
the most striking stabilizing mutations that gives rise to evolution of antibiotic
(TMP) resistance is L28R in Ec DHFR. A 2016 paper PMID: 26929328 provides a detailed
biophysical analysis of this stabilizing rescue mutation and its epistatic effects
on abundance, activity, antibiotic binding etc and possible tradeoffs including
effects of chaperones and proteases.

**Have all data underlying the figures and results presented in the manuscript
been provided?**

Reviewer #1: Yes

Reviewer #2: Yes

Reviewer #3: Yes

PLOS authors have the option to publish the peer review history of their article
(what does this mean?). If published, this will
include your full peer review and any attached files.

If you choose “no”, your identity will remain anonymous but your review may still be
made public.

**Do you want your identity to be public for this peer review?** For
information about this choice, including consent withdrawal, please see our
Privacy Policy.

Reviewer #1: **Yes: **Takeshi Kikuchi

Reviewer #2: **Yes: **George Rose

Reviewer #3: No

---

## [Editor Report · Decision Letter 1]

11 Jul 2022

Dear Dr Varadarajan,

We are pleased to inform you that your manuscript entitled "Mechanistic insights into
global suppressors of protein folding defects" has been editorially accepted for
publication in PLOS Genetics. Congratulations!

Yours sincerely,

Diarmaid Hughes

Associate Editor

PLOS Genetics

Josep Casadesús

Section Editor: Prokaryotic Genetics

PLOS Genetics

Comments from the reviewers (if applicable):

**Data Deposition**

http://datadryad.org/submit?journalID=pgenetics&manu=PGENETICS-D-22-00568R1

**Press Queries**

---

## [Editor Report · Acceptance letter]

22 Aug 2022

PGENETICS-D-22-00568R1 

Mechanistic insights into global suppressors of protein folding defects 

Dear Dr Varadarajan, 

We are pleased to inform you that your manuscript entitled "Mechanistic insights into
global suppressors of protein folding defects" has been formally accepted for
publication in PLOS Genetics! Your manuscript is now with our production department
and you will be notified of the publication date in due course.

With kind regards,

Livia Horvath

PLOS Genetics

On behalf of:
